# Multipurpose self-configuration of programmable photonic circuits

Daniel Pérez-López [1,2 ✉], Aitor López [1], Prometheus DasMahapatra[1,2] & José Capmany [1,2]

Programmable integrated photonic circuits have been called upon to lead a new revolution in information systems by teaming up with high speed digital electronics and in this way, adding unique complementary features supported by their ability to provide bandwidth-unconstrained analog signal processing. Relying on a common hardware implemented by two-dimensional integrated photonic waveguide meshes, they can provide multiple functionalities by suitable programming of their control signals. Scalability, which is essential for increasing functional complexity and integration density, is currently limited by the need to precisely control and configure several hundreds of variables and simultaneously manage multiple configuration actions. Here we propose and experimentally demonstrate two different approaches towards management automation in programmable integrated photonic circuits. These enable the simultaneous handling of circuit self-characterization, auto-routing, self-configuration and optimization. By combining computational optimization and photonics, this work takes an important step towards the realization of high-density and complex integrated programmable photonics.

[1] ITEAM Research Institute, Universitat Politècnica de València, Camino de Vera s/n, 46022 Valencia, Spain. [2] iPronics, Programmable photonics S.L., Universitat Politècnica de València, Camino de Vera s/n, 46022 Valencia, Spain. ✉email: dperez@iteam.upv.es

Integrated photonics is evolving from discrete components to complete application-specific photonic integrated circuits (ASPICs) with ever increasing complexity[1–3]. Although ASPICs have been successfully used to demonstrate many applications, only in a few fields like transceivers and data centers, the fabrication volumes are high enough to compensate for the nonrecurring overhead costs[4,5]. Current time-to-market and time-for-development of photonic integrated circuit (PIC)-based systems is limited by post-fabrication custom processes and by large development periods that range from 12 to 24 months per design-fab-packaging-test iteration[6]. As in electronics, incorporating programmability and reconfigurability into PICs at an increasing pace will enable cost-effective and mass producible, almost-instant development times, negligible up-front non-recurring engineering costs and upgradable components aided by software programming[7–9].

Developing large-scale programmable integrated photonic (PIP) circuits is catalysing considerable research efforts as they are expected to find applications in a myriad of fields, where the flexibility brought by reconfigurability is a unique asset. PIP circuits enable broadband analog processing, which is of particular interest in two scenarios with the considerable impact expected in the short/midterm. The first includes applications where analog processing is required per se, like artificial intelligence, deep learning, and quantum information systems. The second includes applications, such as 5/6 G telecommunications, switching, data center interconnections, hardware acceleration, and sensors, where photonic analog processing can team up with high-speed digital electronic systems to overcome expected limitations arising from the demise of Moore's law.

The processing power of programmable photonic waveguide meshes scales up dramatically with the number of Tunable Basic Units (TBUs) integrating phase actuators in interferometric cells. As shown in Fig. 1a, current state-of-the-art ranges from a few tens to a few hundreds of TBUs per chip[10–18]. However, medium/large-scale complexity PIPs will require many hundreds or even thousands of TBUs, and feasibility will only be possible if soft operations controlling the minimal use of physical resources, optimum circuit placement and configuration, dynamic adaptationm and

self-healing are incorporated. This demands the ability to handle, monitor, and control hundreds or thousands of different variables. Computational optimization methods address the management and control of complex systems[19–21]. Tailoring and adapting these to PIPs can overcome many of their scalability limits. Furthermore, they provide information for compensating nonideal behavior of the components thereby relaxing the need for ideal operation conditions[22].

Here, we develop two multipurpose methods for the self-configuration and optimization of large-scale fault-tolerant PIPs. The first requires data gathered from a periodic, automated pre-characterization routine to configure optical interconnections, delay lines, and optical circuits. The second is based on computational optimization routines that obtain iteratively, without prior pre-characterization, the optimum driving configurations to achieve a targeted functionality. We provide both statistical simulations and experimental verifications to demonstrate, for the first time to the best of our knowledge, the achievement of the targeted circuit configurations, the mitigation of fabrication yield distributed parasitic errors (nonuniform loss and optical crosstalk), and of the dynamic crosstalk arising from thermal and electrical effects through self-configuration. Successful parallel configuration of independent multitasking operations is also demonstrated. Finally, we discuss the impact of the work and comment on the future of this research.

## Results

**Circuit programming based on global algorithms and presets.** To illustrate the proposed multipurpose self-configuration algorithms, we consider the core of a field programmable photonic gate array (FPPGA)[23], as an example of programmable circuit. The FPPGA main architecture shown in Fig. 1b includes the core, several high-performance blocks (HPBs) and optical input/output ports. This configuration enables the synthesis and programming of PICs. The optical core shown in Fig. 1c, is implemented by means of a waveguide mesh. The specific case shown in Fig. 1c is a longitudinally parallel, flat general-purpose hexagonal waveguide mesh[22], although other configurations can

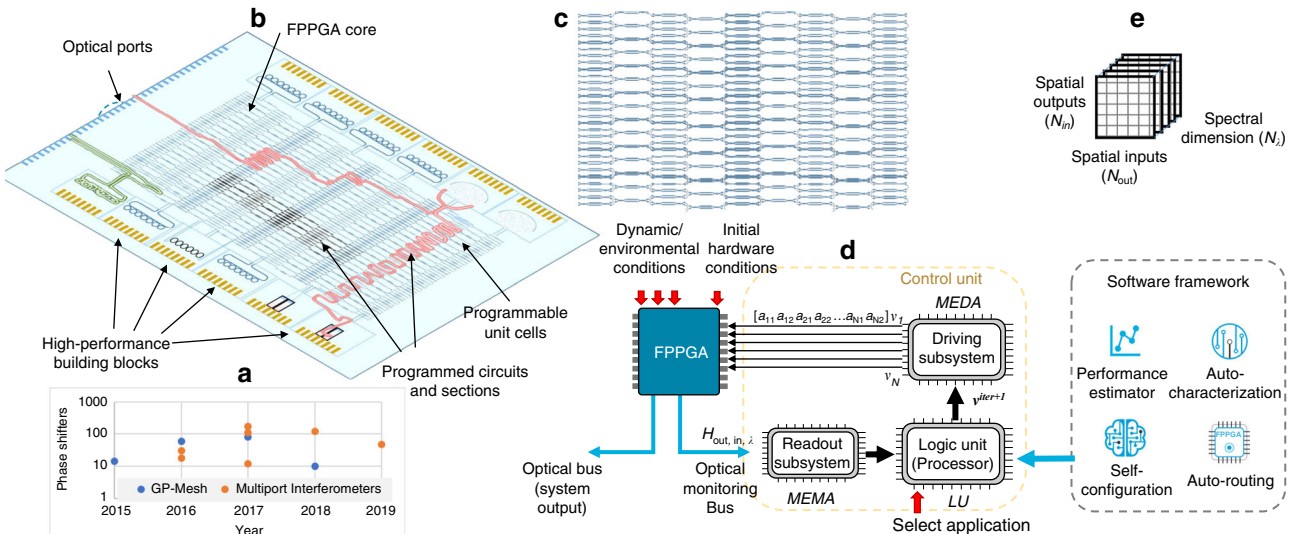

**Fig. 1 Photonic Integrated hardware and control architecture of multipurpose programmable photonic circuits. a** Number of integrated phase shifters in recent waveguide mesh circuits (see Supplementary Note 11), **b** labeled field programmable photonic gate array (FPPGA) architecture, including a waveguide mesh core and high-performance building blocks, **c** FPPGA core employing a longitudinally parallel hexagonal waveguide mesh interconnection topology[22], **d** electronic control subsystem, signals and software procedures to control the programmable photonic integrated circuit, **e** data array of the full scattering matrix of the FPPGA core, including input and output spatial ports and the optical spectral dimension. GP general-purpose, MEMA multichannel electrical monitoring array, MEDA multichannel electrical driving array, LU logic unit.

be considered[16,24]. Figure 1b illustrates a given operational scenario where two circuits (in red and green) are configured in the FPPGA.

Control and configuration protocols of the FPPGA have not been demonstrated to date. They can be potentially classified into different types, depending on the scope, the control strategy, and the hardware requirements in terms of number of optical readouts and their position in the circuit. In this work, all the reported methods employ the closed loop control system illustrated in Fig. 1d. It includes an electrical IC to read a signal proportional to the optical power at every external optical port in the circuit ($N_p$), an electronic processor to run the algorithms and a driving circuitry to configure the phases at every TBU, described by the vector **v**. For most algorithms, we define an operation as the completion of a cycle involving the configuration of $v$, and the extraction of the full scattering matrix of the aforementioned circuit. In most of examples, only a portion of the scattering matrix is required. (For additional details, see "Methods" and Supplementary Note 1).

The set of algorithms developed in this section require a prior knowledge of data, regarding the mesh architecture and full bias calibration information of every TBU behavior. The information involving the architecture itself, includes the interconnection scheme, and the physical features that define the TBUs, such as their basic unit length (BUL) and basic unit delay[25]. The information regarding the full control of every TBU includes the computation of the nonideal passive offsets of the phase of each TBU, the calibration curve (tuning response) of each phase shifter, estimated insertion loss, power consumption[26,27], and the tuning crosstalk matrix that characterizes the undesired coupled effects of neighboring phase actuators. These requirements can be obtained by means of the periodic application of self-characterization routines based on iterative maximization and minimization methods[28], and regression-based approximations (see implementation and detailed analysis in Supplementary Note 2).

Auto-routing and pathfinding algorithms are software routines capable of finding optimum optical paths or interconnections between any two connections of a FPPGA, with respect to a set of pre-defined figures of merit and the data gathered from the self-characterization stage. They can also be applied to define circuits or interconnections between programmed components and external HPBs, as shown in Fig. 1b. The circuit architecture is mapped into a graph (from graph theory), where optical nodes between programmable unit cells define the graph's vertex and their connections define the graph's edges. It is then possible to apply graph-based optimization algorithms to configure the targeted circuit. Each edge or interconnection is loaded with the associated key limiting factors or features ($f_i$) gathered during the characterization stage in the form of a weighted sum called transmission distance (TD), as described in Eq. (1), and the accumulated penalties are then considered by the optimizer. To perform the auto-routing task, a shortest path tree algorithm with restrictions can be employed[26,27]. More information can be found in Supplementary Note 3.

$$TD_{xy} = c_1 \times f_1 + c_2 \times f_2 + c_3 \times f_3 + \ldots \quad (1)$$

The auto-routing algorithm can be combined with the self-characterization routines described in Supplementary Note 2 to dynamically configure the FPPGA core. Following the scheme described in Fig. 1d, our measurement setup consists of a multichannel electronic driver array subsystem based on a tabletop multichannel current source along with an Optical Spectrum Analyzer as an optical monitor, with custom routines in Python being run on a standard personal computer, thus completing the

entire feedback loop along with a processing unit. We employed the setup in a 7-hexagonal waveguide mesh with 30 TBUs with the circuit being reported in ref. [17].

To conduct the experiment, as shown in Fig. 2a, we first employed the self-characterization cycle to calibrate the responses of every phase actuator and TBU, and their power consumptions. Then, we used this information as input for the auto-routing algorithm to set up seven different circuit implementations, referred in Fig. 2b as configs. 1–7: a 10-TBU optical ring resonator (ORR), a 4-TBU imbalanced MZI, a simultaneous combination of a 6-TBU ORR working in parallel with a 2-TBU imbalanced MZI, a 6-TBU ORR, a second-order coupled resonator optical waveguide with cavity length of 6-TBU, a simultaneous combination of two delay line channels of 6 and 5 TBUs, and a 12-TBU ORR. Translating each circuit configuration to an array $v$, including the required electrical current value for each phase actuator in the arrangement and loading each at a time to the processor allows us to demonstrate the first dynamic reconfiguration of a multipurpose waveguide mesh arrangement. Figure 2b illustrates how the waveguide mesh arrangement functionality evolves over time. The precise spectral performance can be found in the lower insets. The tuning steps are limited to jumps of 5 mA to prevent undesired overshoots, to protect the circuit and to better illustrate the dynamic response between configurations. Each step can be done in <1 s, limited by the hardware of the control system employed and its USB connections, as well as the Serial Peripheral Interface.

**Circuit programming based on computational optimization.** Algorithms requiring a previous knowledge and characterization routines can be employed, as shown above to control and program FPPGAs, and waveguide meshes. However, the periodic execution of self-characterization routines can be time and resource intensive when scaling FPPGA cores to a large-scale number of programmable units. Information gathering requirements for smart control techniques, addressing nonuniform loss distributions over the circuit, dynamic tuning crosstalk between actuators and optical crosstalk due to imperfect design and fabrication can render these methods inefficient. Ideally, programming, supervision, and control should be achieved without the need for prior knowledge and characterization.

Here, we propose, apply, and demonstrate a general scheme of computational optimization methods to program the FPPGA core, as shown in Fig. 1b. We illustrate it with two configuration examples, an *all-cross* TBU interconnection configuration and the automatic definition of optical filters (a third example dealing with a $1 \times 8$ beamsplitter can be found in Supplementary Note 5). In every case, we consider the control scheme architecture in Fig. 1d. However, as opposed to the previous approach, this method does not require prior information (pre-characterization). This time, we assume that the FPPGA core is a black box that returns the scattering matrix datasheet as a function of the chip´s passive and dynamic conditions, and the electrical driving. Then, the optical readout system extracts a portion of the scattering matrix and closes the feedback loop of the optimization system. This portion of the extracted matrix is employed to compute the cost function that is minimized when the targeted application is achieved. Each computational optimization algorithm employs different strategies to decide the new driving configuration to get closer to the targeted final operation through the minimization of a customized cost function. Thus, the definition of a cost function specific for each application is a non-straightforward task and different cost functions can be employed to define the same application, thereby achieving different convergence rates and compromising the success of the

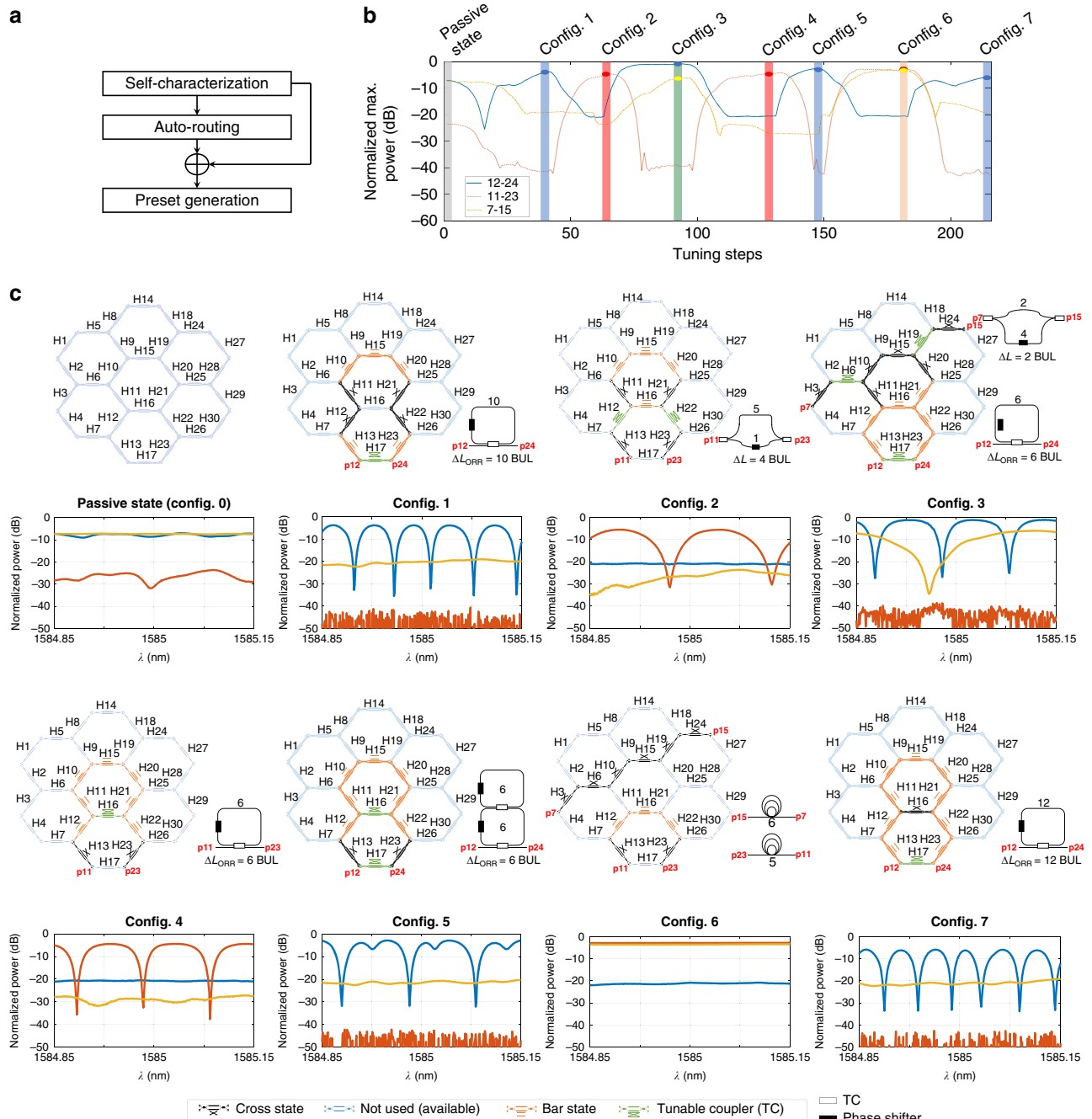

**Fig. 2 Experimental results for sequential circuit programing using auto-routing and prior-knowledge-based algorithms.** The algorithms are applied to a 30-Tunable Basic Unit hexagonal waveguide mesh with measured normalized maximum optical powers at channels 12–24, 11–23, and 7–17. **a** Workflow of the experiment following a self-characterization routine, the auto-routing algorithm, and the generation of presets, **b** dynamic configuration illustrated by the evolution of the normalized maximum optical power versus tuning steps with maximum current step change of 5 mA allowed per phase actuators for the three optical channels, **c** the waveguide mesh arrangement with the relevant unit cells configured in passive, cross, bar, or tunable coupling states (up), and the normalized spectral response measured for each circuit configuration (down) for the following seven configurations: config 0: passive state, config. 1: optical ring resonator (ORR) defined by 10 basic unit lengths (BULs), config. 2: Mach–Zehnder interferometer (MZI; 4 BUL), config. 3: MZI (2-BUL) and ORR (6 BUL), config. 4: ORR (6 BUL), config. 5: coupled resonator optical waveguide (CROW; 6 BUL), config. 6: delay line (6 BUL) and ORR (6 BUL), and config. 7: ORR (12 BUL). Traces are normalized to a straight waveguide with coupling and propagation loss of 22 dB.

algorithm. In our results, we show both the sensitivity to different cost functions definitions, to their specific hyperparameters and the statistical behavior of the proposed methods.

Optimization methods can be classified according to different criteria: global and local search algorithms depending on their suitability to find the global maximum or their ability to converge rapidly to their closest minimum point; derivative and nonderivative methods depending on whether the computation of the gradient is employed for each iteration or not; deterministic or stochastic methods depending on the inclusion of random variables during the procedure; and individual or population approaches, depending on the number of search points employed during each iteration. In order to test the application of optimization methods for the control of large-scale programmable

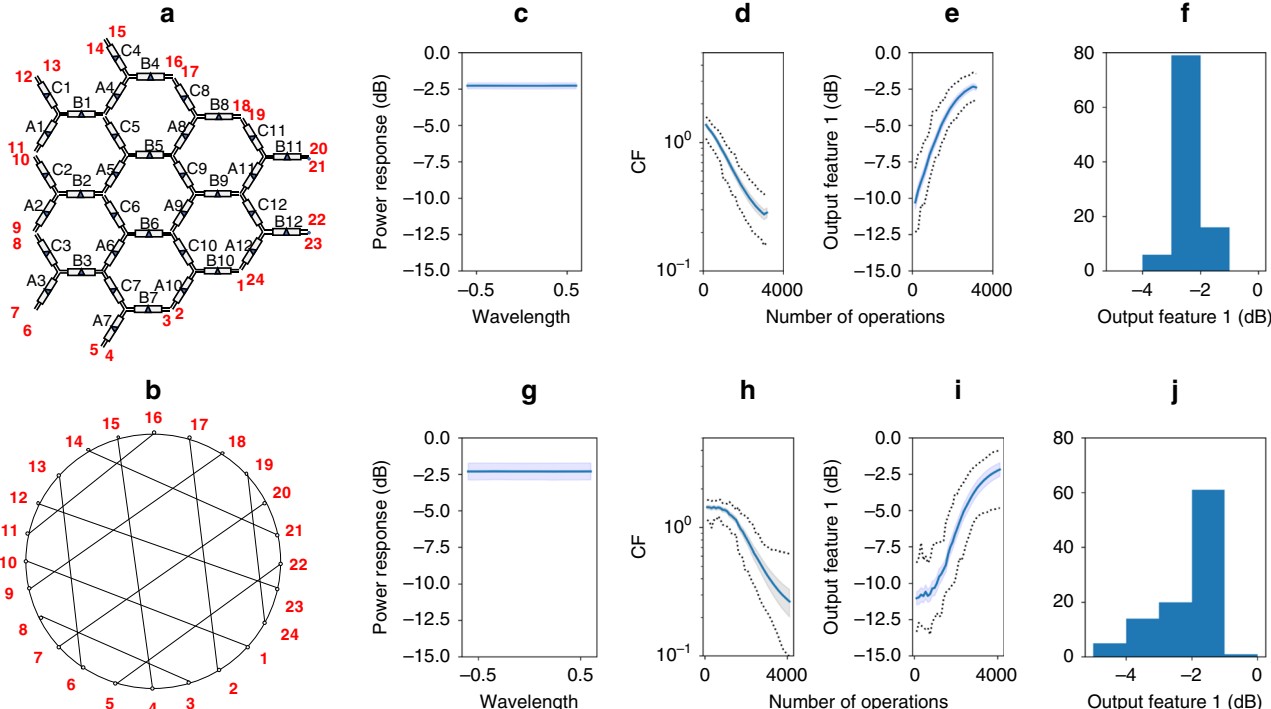

**Fig. 3 Numerical results for the self-configuring of an all-cross function in a 36 TB hexagonal waveguide mesh.** Note that the process involves using computational optimization methods and no prior structure testing or calibration is employed. **a** Labeled schematic of the waveguide mesh arrangement under test with both TBU (black) and port (red) labels. **b** Black box system with the targeted performance, where the system performs an optical routing between channels defined by the port pairs 12–23, 14–21, 10–1, 8–3, 13–6, 15–4, 17–2, 19–24, and 11–16, 9–18, 7–20, 5–22. Note: they represent direct connections without crossings or splitting. All-cross function statistical results for fixed hyperparameter selection with **c–f** genetic algorithm and **g–j** particle swarm optimization for $CF^{all-cross}_1$. From left to right: **c**, **g** spectral response versus wavelength normalized to the basic unit delay (BUD)[25], **d**, **h** evolution of the average (solid), maximum and minimum (dotted), and standard deviation (shaded) cost function, and **e**, **i** output feature(OF) and **f**, **j** histogram of the last operation (OF1: average of normalized output channels power of the beamsplitters, the datasheet is composed of 100 independent experiments with different arbitrary waveguide mesh initial conditions.

PICs, we employ first a performance estimator based on an induction method[29]. This allows us, in further sections, to introduce nonideal effects and to perform a progressive analysis. In addition, some of the results are experimentally verified with a real waveguide mesh PIC in the final section. See "Methods", and Supplementary Note 1 and 4 for more details.

*Self-configuration of all-cross* function aims to configure every phase actuator so that all TBUs in the waveguide mesh are in cross-state. This function is particularly interesting for both calibration and characterization, as well as for setting an optimized initial point for some of the optimization methods described in this work and enhance their convergence. The cost function is just an example of a broad range of applications involving arbitrary routing of the optical signals between the optical ports.

In this example, illustrated in Fig. 3, the optical channels are described by the following port pairs: 12–23, 14–21, 10–1, 8–3, 13–6, 15–4, 17–2, 19–24, and 11–16, 9–18, 7–20, 5–22. Note that the last set of four pairs of ports is redundant and could be deleted from the cost function, as they do not incorporate additional TBUs. The new cost function is described as a weighted sum as in Eq. (1), with the following coefficient and feature definition. (See Supplementary Note 6 for a full parameter description and definitions):

$$CF^{allcross}_1 = \begin{cases} c_1 = -1, \\ f_1 = \frac{1}{N} \sum_{1}^{chs.} (\log(|H_{chs.}|)), \end{cases} \quad (2)$$

where $N$ is the number of optical channels incorporated in the optimization process and $|H_{chs.}|$, the maximum absolute value of

the intensity of the electrical field measured for each optical channel listed before. To perform a qualitative analysis of the achieved performance, we monitor and define the output feature 1 as the average power transmission response in the targeted optical channels defining the all-cross operation. It is expressed in logarithmic units.

We performed the self-configuration process based on an evolutionary genetic algorithm (GA), a particle swarm optimization (PSO), and gradient descent with momentum. For each case, we performed a grid search with all the hyperparameters involved in the optimization to demonstrate their impact on the efficiency of the configuration process (see "Methods" and Supplementary Note 6). The sensitivity to the hyperparameters selection that configure the algorithms is mitigated through the use of adaptative values allowing the scheduling of the exploration and exploitation capabilities of the self-configuration task. It results in a robust statistical behavior, once we are close to the optimal set of hyperparameters that configure the algorithm. As shown in Fig. 3, we repeated 100 times the self-configuration routine and obtained an average error better than 3-dB in the 95% and 76.66% of cases for the GA and the PSO, respectively, in <3000 and 4000 operations. The PSO hyperparameters configuration included an adaptative inertia from 0.9 to 0.35, 108 particles, cognitive 0.7, and social 1.8. The GA employed a population of 144 particles, where the 20% mating and a 50% muting. A decay is scheduled for the muting weight. Finally, we performed a gradient descent with momentum approach, revealing that in the best case, it is possible to complete the task in 750 operations with a learning rate of 50 and a momentum coefficient of 0.5. A similar

application example, the $1 \times 8$ optical beamsplitter is covered in the Supplementary Note 6.

*Self-configuration of optical filters* function enables programming the waveguide mesh arrangement to suppress a given spectral band while maintaining the minimum losses in the passband. Self-configuration automatizes this process and involves the selection between thousands of parameters, dealing with the mitigation of nonideal effects like optical crosstalk, tuning crosstalk, providing power consumption and reducing the hardware resources (TBUs) to perform the targeted function, and the optical losses. For the definition of the cost function, it is possible to consider many features, such as the insertion loss of the passband, the extinction ratio of the filter, the roll off, the optical power at the nontargeted ports, etc. In addition, one can define the targeted spectral mask of the filter and define a cost function feature considering the error between the obtained trace and the mask at each iteration. In this example, we show the self-reconfiguration capability employing only the mean square error in the spectral mask, although multiple objectives could be added to the cost function:

$$\text{CF}_1^{\text{O.Filter}} = \begin{cases} c_1 = 1, \\ f_1 = \frac{1}{N_\lambda} \sum_\lambda \left( M_\lambda - 20 \log_{10}\left( \left| S_{4,1}(\lambda) \right| \right) \right)^2, \end{cases} \quad (3)$$

where $N_\lambda$ is the number of wavelength points, $M_\lambda$ is the value of the spectral mask at each wavelength point, and $S_{4,1}$ is the value of the scattering matrix at the optical channel defined by ports 4 and 1 at a given wavelength. Note that this operation is equivalent to the average of the distance between the mask and the measured trace. See Fig. 3 for the location of the ports. To demonstrate the performance of the self-configuration method, we defined a set of different spectral maks (see Fig. 4).

To compare between methods and cost functions, we analyzed also the evolution of two output features that account for the insertion loss in the passband ($OF_1$) and the extinction ratio of the filter ($OF_2$). We tested three approaches based on GA, PSO, and gradient descent with momentum. For each approach, we tested the sensitivity of the hyperparameters configuring the algorithms employing, for all cases, a grid search for the hyperparameters of a mask describing a 2-BUL interferometric free spectral range (FSR), illustrated in Fig. 4a–d. The GA–PSO approaches provide good hyperparameter combinations for convergence, whereas the gradient descent approach offered less and sometimes nonexistent convergence success. For the GA and PSO, the use of adaptative rates (mainly for mutation and inertia, respectively) provides better results in terms of convergence to an optimum configuration and shows less sensitivity to their selection. (See Supplementary Note 7 for extended results). In particular, we illustrate the behavior of a PSO based self-configuration maintaining a fix set of hyperparameters (adaptative linear inertia: 0.9–0.35 in 1000 iterations, cognitive: 0.7, social: 1.8). The figure plots different spectral masks, the average spectral response, standard deviation, and approximated 95% confidence interval, assuming gaussian distributions after 4000 operations. Results demonstrate that the self-configuring routine tunes the phase actuators to achieve the targeted mask successfully, thus creating the required interferometric patterns in the waveguide mesh arrangement. A broad range of filters with FSRs comprising of different TBU numbers have been considered in the trials. Also, a wide range of passband and stopband bandwidths, and extinction ratios are tested. We observed that some of the structures require a larger-scale mesh arrangement to provide the targeted higher-order filter masks, although the obtained results are by themselves remarkable (see Fig. 4e–x).

*Self-configuration managing nonideal components:* self-configuration in the presence of dynamic tuning crosstalk increases the complexity of the task due to the undesired thermal coupling between the phase actuators. To test the behavior of the proposed methods, we performed two statistical tests, with and without the presence of thermal crosstalk. In the latter, we loaded the performance model[29] with a severe crosstalk matrix, where the crosstalk coefficients are obtained from a uniform distribution from 0 to 5% (see "Methods"). As shown in Fig. 4y–bb, the self-configuration routine considers the dynamic tuning crosstalk effect during the optimization procedure, thus obtaining the targeted spectral shape. We ran a second example with a different spectral mask and crosstalk from a uniform distribution up to 10%. Results are plotted in Fig. 4cc–ff. These scenarios are by far, more challenging than the ones experienced by a real system, where the crosstalk ranges from 2 to 3% and is decreasing as a function of the distance[17,30,31]. These results open the path for employing waveguide mesh arrangements with much higher integration density, reducing the distance between components[22]. See Supplementary Note 7, for extended results.

Self-configuration also enables fault-tolerant and self-healing circuits. While fabrication and component yields can completely discard a whole die in application-specific PICs, waveguide mesh arrangements offer potential fault-tolerant and self-healing capabilities. This advantage comes from the repetition, interconnection and cooperation of simple components in the mesh that enable the use of alternative circuit paths, when some parts of the circuit are damaged. To illustrate a demonstration example, we configured a filter specified by the spectral mask, appearing in Fig. 4gg–jj, where we decreased the performance of TBUs with a high probability of use (A6, B6, and C6, see Fig. 3a) by imposing 30 dB insertion loss to each one. After running the statistical test, the self-configuration process is able to provide an alternative layout in the available mesh space and maintain, most notably, the demanded response (for extended results, see Supplementary Note 7).

A final experiment, illustrated in Fig. 4kk–nn, demonstrated the ability of the self-configuration algorithm to program simultaneously two circuits. It employed two spectral masks with different characteristics and considered ports 1–4 (blue) and ports 23–21 (red). This result opens the path for the automated configuration of multitasking operations in multipurpose programmable PICs.

**Experimental demonstration of self-configuration methods**. Finally, as illustrated in Fig. 5, we programmed several interferometric circuits experimentally in the 30-TBU waveguide mesh using PSO algorithm, following the same methodology and cost function as in the simulated examples. The spectral masks were obtained using previously obtained current presets. We limited the number of variables to be optimized to 13, as it was the number of current sources available for the experiment. In <416 operations, the algorithm returns the targeted spectral mask with average MSE errors of 0.542, 0.702, and 4.139 for configs. 1, 2, and 3, respectively. For additional details of the experimental setup, see Supplementary Note 8. For additional configurations, details of the architecture, optimization methods, and extended results, see Supplementary Note 9.

## Discussion
Most reported state-of-the-art reconfigurable photonic circuits are configured manually or in a semiautomated manner by means of open-loop control, utilizing look-up tables that includes the pre-characterization of each tuning element and inter-element crosstalk[17,32–36]. Although a significant number of proof-of-

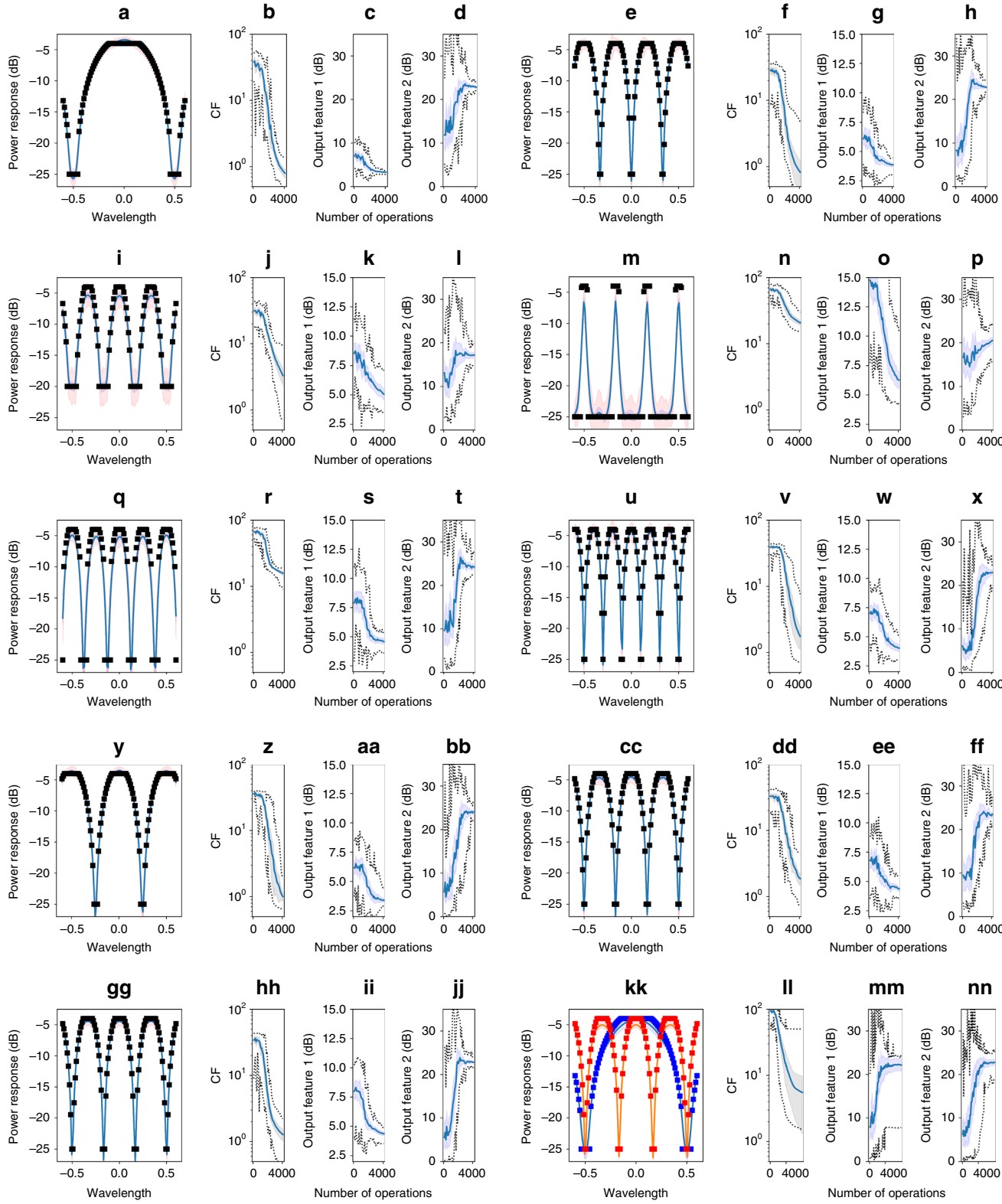

concepts and standalone components have been demonstrated, this approach does not scale efficiently with the number of tuning elements in complex large-scale circuits, including hundreds or thousands of components and actuators. These circuits demand both a dedicated control system and a software layer to automate their configurations for two reasons. First, the complexity of the targeted operations and nonideal components would lead to configurations that cannot be predicted in terms of pre-

characterized data or human experience. Secondly, agile programmable systems require a fast and reliable feedback mechanism to exploit their reconfiguration capabilities.

The former is only one of the many challenges to address in developing large-scale programmable photonic circuits. Limitations connected to fabrication defects, design deviations, passive parasitic effects, power consumption, thermal crosstalk, and robustness of phase shifters must be considered, in conjunction

**Fig. 4 Examples of statistical results for the self-configuration performance evolution of optical filter programming.** The process employs different spectral masks using particle swarm optimization algorithm with fixed hyperparameter selection—inertia lineal from 0.9 to 0.35 in 1000 operations—and cognitive: 0.7, and social 1.8 and 108 particles—employing the cost function defined in Eq. (3), $CF^{O.Filter1}$ and ports 1–4 represented in Fig. 3. **a–x** For each four-plot image group, we evaluate from left to right: (1) self-configuration of different spectral masks (squares) without tuning crosstalk and with nonuniform loss, representing the spectral response (solid) with wavelength normalized to unit cell delay, (2) the evolution of the average (solid), maximum and minimum (dotted), and standard deviation (shaded) cost function, and (3) output feature (OF) values. OF1: insertion loss of the passband (dB), OF2: extinction ratio (dB). Each datasheet is composed of 30 independent experiments with different waveguide mesh initial conditions. Note that each example request different periodicity and shapes, **y–bb** self-configuration of different spectral masks (squares) and evolution of the performance with nonuniform unit cell loss and tuning cross talk up to 5%, **cc–ff** self-configuration of different spectral masks (squares) and evolution of the performance with nonuniform unit cell loss and tuning cross talk up to 10%, **gg–jj** self-configuration of different spectral masks (squares) and evolution of the performance with nonuniform unit cell loss and "damaged" unit cells A6, B6, and C6 (see Fig. 3 for reference), **kk–nn** self-configuration of two spectral masks simultaneously employing ports 1–4 (blue) and ports 23–21 (red). Spectral resonses (solid) versus wavelength is always normalized to the basic unit delay (BUD).

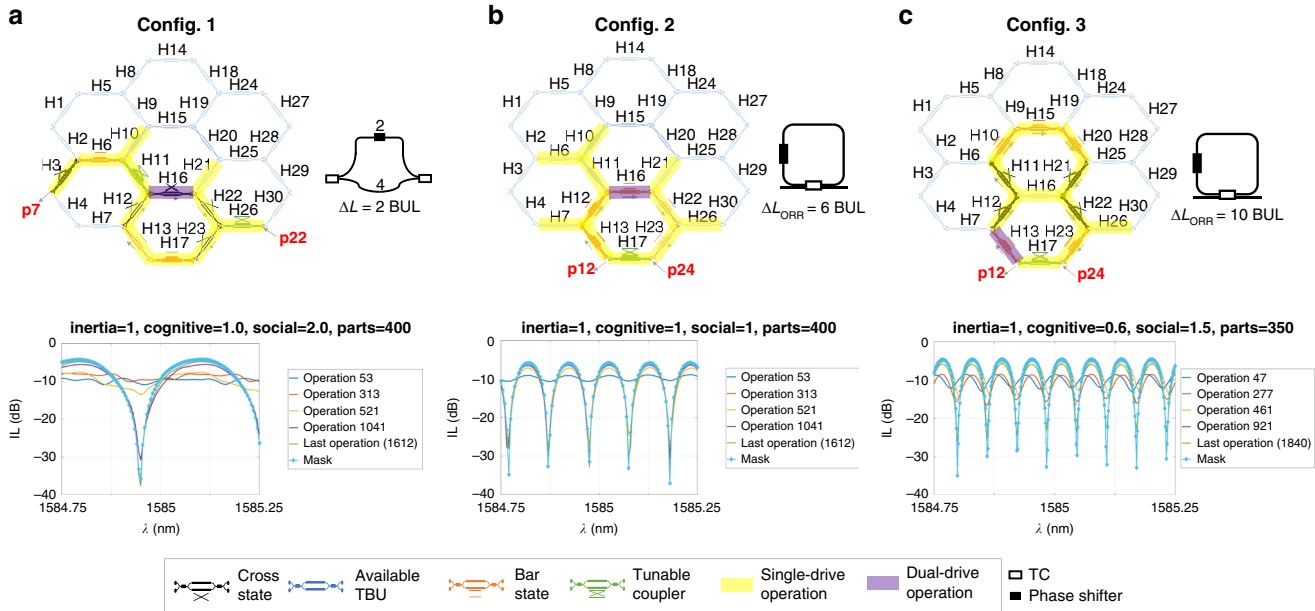

**Fig. 5 Experimental realizations of several interferometric circuits in a waveguide mesh using particle swarm optimization algorithm. a** Labeled schematic of the electrically driven units for a targeted 2-Tunable Basic Unit (TBU) imbalanced Mach–Zehnder interferometer (MZI) and evolution of the spectral traces employing 101 points for a set of selected operations. **b** Labeled schematic of the electrically driven units, for a targeted 6-TBU optical ring resonator (ORR) response and evolution of the spectral traces employing 301 points for a set of selected operations. **c** Labeled schematic of the electrically driven units, for a targeted 10-TBU ORR and evolution of the spectral traces employing 501 points for a set of selected operations. The average MSE obtained for each of these configurations after the end of the process were of 0.542, 0.702, and 4.139, with ER of around 25 dB for the 2-TBU MZI and the 6-TBU ORR, and 20 dB for the 10-TBU ORR. The overall number of operations in each experiment was of 1612 for **a** and **b**, and 1840 for **c**.

with others connected to reconfiguration speed and enhanced convergence of automated functions. Due to space restrictions, we specifically cover these in Supplementary Note 10. In addition, a significant challenge resides in the efficient management of control circuit complexity, the interfacing of a large number of electronic signals and the associated large number of variables and equations to be solved. Although different in application, recent published chips with similar requirements integrate a great number of optical power monitors and phase shifters. For example, two recent demonstrations integrate 900 optical power monitors and 448 phase shifters[37] and 1024 phase shifters[38], respectively, in footprints <300 mm². 

Equally important is the correct management of the complexity of optimization algorithms and circuit programming. Proposals have been reported to program PICs targeting specific applications. Examples of this include the configuration of multiport interferometers based on the individual control of tunable couplers and switches[12,39,40], automated path scheduling and assessment of a complex wavelength-and-space-switch circuit[41], automated bank filters[42–46], reconfigurable all-pass filters[47] and

beamforming networks[48]. Large-scale and very large-scale PICs will find it difficult to rely on local feedback loops for controlling each tunable unit due to the excessive overhead caused by monitors and electronic access tracks.

The methods reported in this paper open a new direction toward an automated and reliable programming of complex multipurpose PICs. The best of two worlds can be exploited combining customized routines requiring data from automated pre-characterization routines and configuration strategies based on computational optimization methods. The first set provides a fast configuration response but does not solve tuning-based crosstalk problem, which is exacerbated in large-scale high-density circuits. In addition, the self-characterization routine needs to be applied periodically in order to compensate environmental variations. The second set combines creation of configuration-specific cost function definitions and optimization routines to self-configure complex circuits. Although we have provided a first demonstration of self-configuration functionality, results can be improved in terms of convergence speed. Further research is required to identify solutions incorporating more advanced optimization routines, cost function tailoring, and

combination and scheduling of exploratory and local search methods, and machine-learning techniques[21]. Finally, the inclusion of multi-objective figure of merit opens the door to programming protocols embracing targeted performance, as well as additional features, such as power consumption and savings in the use-of-components. These can be optimized with multi-objective optimizers[49]. Regarding the configuration speed, an operation cycle is mainly limited by the time required to measure the desired portion of the scattering matrix. If this time is ~100 ms for 21 elements of the scattering matrix, a full reconfiguration process of 3000 operations would last ~5 min. These metrics can be improved enhancing the convergence and by combining both proposed configuration approaches to reduce the number of variables during the optimization process. Further details in this are in Supplementary Note 10. In this respect, the self-healing, fault-tolerant attributes of the reported work are expected to impact the methodology leading to the design and interaction with high integration density flexible and versatile multipurpose programmable PICs.

All in all, advances in photonic integration technologies, encapsulation, and novel designs and architectures pave the way for the development of a new class of multipurpose programmable PICs. Including hundreds or thousands phase actuators over a massively coupled arrangement of waveguides, these circuits potentially provide a flexible hardware framework to perform multiple and versatile applications. However, the absence of effective control and programming strategies limited the scalability of current demonstrations. In this work, we presented a control architecture and a set of control strategies to perform fault-tolerant self-configuration of the circuit to perform a targeted task. These algorithms are divided by configuration methods requiring pre-characterization routines, also presented, and advanced optimization methods that not only avoid this requirement but also overcome the presence of nonideal components with nonhomogeneous loss distribution, power consumptions, phase offsets, optical crosstalk, and tuning crosstalk. We proposed and developed the use of self-configuration routines based on stochastic population-based methods, such as GAs PSO for three applications, an all-cross router, a beamsplitter, and an optical filter configured for a wide variety of spectral masks. By combining computational optimization and photonics, this work makes important step toward a new paradigm in photonic integration: programmable photonics.

## Methods

**Architecture, variable space, and cost function definitions**. The self-configuration algorithms proposed in this paper rely on the application and customization of optimization routines employed in a wide range of application fields. Optimization deals with the task of finding the optimal values for the variables of a system to maximize or minimize its output. An ideal optimizer should avoid stacking in local optima and explore efficiently the variable's space to find the global optimum point, as well as converge efficiently when this is found.

In most of the applications and algorithms demonstrated in this paper, the system to be optimized is the general-purpose PIC. We consider the PIC as a black box whose response is given by the full scattering matrix of the circuit. The scattering matrix contains the spectral response of every optical port combination. Although a real system contains amplitude and phase response, we employ the overall amplitude response in our current methods. The overall system is modified through the application of a set of electrical signals that modify the optical properties locally in the circuit. This set of variables is defined by the vector **v**. When dealing with a real system, this vector can represent the electrical signal feeding each phase actuator or photonic actuator in general, whereas we employ phase shifts for the performance estimator that includes a model of the nonideal performance of every component[29].

In this work, we apply both stochastic and derivative optimization techniques. In all cases, the optical system response is given by the settings applied to each phase actuator ($v$), the initial hardware conditions such as nonuniform loss distributions per TBU in the system, as well as the passive conditions ($C_p$). These nonideal effects coming from fabrication deviations of the waveguide geometry and environmental conditions modify the system performance, are considered to be unknown and random, and are selected from a normal distribution of mean 0.15

dB and std of 0.05 dB, and from a normal distribution between 0 and $2\pi$, respectively, unless otherwise specified. For each wavelength, we obtain the scattering matrix of the circuit that represents the optical response for every input and output port combination as $S_{o,i}$.

$$S(v, \lambda, C_p) = \begin{pmatrix} S_{11} & S_{12} & \cdots & S_{1P} \\ S_{21} & \ddots & & \vdots \\ \vdots & & \ddots & \vdots \\ S_{P1} & \cdots & \cdots & S_{PP} \end{pmatrix}_\lambda = f(v, \lambda, C_p). \quad (4)$$

With the scattering matrix information, or a portion of it, we compute the cost function that is dependent of the application to be optimized. A process of cost function engineering requires the search of a function that is minimized when the targeted application is achieved. Once the cost function is computed, the optimization algorithm computes the next vector of variables for the driving system. Thus, the optimizers deal with finding the optimum values for the individual phase actuators in the circuit to minimize a cost function and get the desired response of the optical processor, even in the presence of nonideal conditions. The full cycle of driving, and monitoring is defined as an operation along this paper. The self-configuration algorithms proposed require the computation of a certain number of operations per iteration, so we rescale to number of operations to compare their performance (for additional details see Supplementary Note 1).

*Understanding, simulating, and modeling thermal crosstalk*: when tuning one phase actuator, for example a thermo-optic actuator, the physical effect causing the tuning in the desired waveguide can spread to the neighboring waveguides producing an undesired tuning effect. In the worst cases, even at distances >10 mm the tuning crosstalk effect can be appreciated[17,50]. The tuning crosstalk can be modeled by a constant that reflects the percentage of phase shift occurred in the nontargeted waveguide compared to the one experienced by the targeted waveguide. Simulations and experimental works result in a crosstalk coefficient between 1 and 3% at several hundreds of micrometers[30,31]. If extended to a system with multiple phase shifters, this effect can be modeled by a system of equations relating the effective phase shifts with the phase shifts set by the algorithm or the user.

$$\Delta\phi_{\text{effective}} = \begin{pmatrix} 1 & CT_{12} & \cdots & \cdots & CT_{1N} \\ CT_{21} & 1 & & & CT_{2N} \\ CT_{31} & & 1 & & \vdots \\ \vdots & & & \ddots & CT_{1N-1} \\ CT_{N1} & CT_{N2} & \cdots & \cdots & 1 \end{pmatrix} \begin{pmatrix} \Delta\phi_1 \\ \Delta\phi_2 \\ \vdots \\ \\ \Delta\phi_N \end{pmatrix}. \quad (5)$$

**Computational optimization methods**. *Genetic algorithm*, also known as evolutionary algorithms, resemble natural selection and reproduction processes governed by rules that assure the survival of the fittest individuals in large populations. Individuals (points) are associated with identity genes that define a fitness measure (objective function value). A set of individuals form a population, which adapts and mutates following probabilistic rules that utilize the fitness function. In this case, our individuals are defined by $v$. For each generation, we select a percentage of the population that achieves the lowest cost function for *matting* between them and apply a mutation value that will decrease exponentially over each generation. In general, is classified as a global-search algorithm.

*Particle swarm algorithm* is a population-based algorithm that maintains at each iteration a swarm of particles (set of points) with a velocity vector associated with each particle. At each iteration, it generates a new set of particles from the previous swarm combining random and inherited parameters (inertia, cognition, and social). It is typically classified as a global-search algorithm.

Both of them are nonderivative methods. For additional details and derivative methods included, see Supplementary Note 4.

## Data availability

All data are available from the corresponding author upon reasonable request.

## Code availability

All codes are available from the corresponding author upon reasonable request.

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

## Acknowledgements

D.P.L. acknowledges funding through the Spanish MINECO Juan de la Cierva program. J.C. acknowledges funding from the ERC Advanced Grant ERC-ADG-2016-741415 UMWP-Chip and ERC-2019-POC-859927. Authors also acknowledge funding from Future MWP technologies and applications PROMETEO/2017/103, Advanced Instrumentation for World Class Microwave Photonics Research IDIFEDER/2018/031, EUIMWP CA16220, Infraestructura para caracterización de Chips Fotónicos EQC2018-004683-P.

## Author contributions

D.P.L. and A.L. conceived the control and configuration routines. D.P.L. and J.C. conceived the processor design. D.P.L. designed the chip. D.P.L. and A.L. conceived the experiments, and performed the simulations and measurements. D.P.L. and P.D. supervised the experiments. D.P.L., A.L., and J.C. analyzed the data and wrote the paper. D.P.L., P.D., and J.C. managed, coordinated, and supervised the project.

## Competing interests

The authors declare no competing interests.
