## [Peer Review File · Nature Communications]

REVIEWER COMMENTS

Reviewer #1 (Remarks to the Author):

Programmable photonic circuits are a highly interesting topic in the field of photonics, particular for signal processing. It promises to bring implementation of signal processors to a new generation by combining digital electronics and photonics. This practice explores the capabilities of photonics towards the complexity level of CPUs which is currently a great challenge in science and it is expected to benefit a variety of applications, e.g. AI, 5G, datacenter, etc., with significant improvement in signal processing bandwidth, latency, and power consumption which are considered the key features for next generation commercial signal processors. Therefore, the effort in this field is of high value for both academia and industry. The work shows in this paper is an important step in the path of implementing such signal processors by addressing the challenge of automatic control for scalable circuit complexity. I agree that the results here verify authors' concept and are interesting for the general readers, particularly in terms of control and optimization algorithms, encouraging deeper study and discussions in the field. I recommend acceptance for publication, however, I suggest the authors providing more discussion on a few points, if alright.

1. The photonic circuit in demonstration measures 300mm². When you further scale it up for higher circuit complexity. How do you deal with the uniformity and risk of defect in fabrication from a yield perspective? Would the expected yield still justify its use, compared with mature digital processors in a multi-core/parallel scheme?

2. The basic component performance metrics, like waveguide loss, back reflection, nonlinear effect, coupler function accuracy, heater maximum current, life time, and possible electromigration, of the current silicon technology fundamentally limits the complexity and functions of such circuits. This is a first of all physics level challenge before talking about applications. What is your view on this?

3. Such circuits are able to perform signal processing with a fixed configuration at a time. Do you envision to use such circuits to performance computing with parameter updating speed at bit-rate level?

4. We dont have optical memory at the moment and still need to use external digital memory for a complete signal processing system. The implementation of communication adds power consumption, uses E/O devices, and may also limit overall system processing speed. Considering this, how do you justify the advantage of using photonic processors?

5. The operation of such photonic circuits would need particular driver circuits, cooling system, and may require light source as well as amplifiers. When would such photonic solutions have a definite advantage over the digital processor solutions in terms of speed and power consumption at the level of a stand-alone signal processing module? Is it possible to provide a side-by-side comparison?

6. About self-characterization, when the circuit complexity is high, how do you get access to each of the TBUs in the circuits and measure the properties? When you have a large number of unknowns and you would need as many equations to solve all of them. In practice, the number of TBUs in a 2D array would be much larger than the number of optical I/Os, if I understand it right. How would this work out? and would the solution still apply when the circuit complexity change again?

7. About optimization algorithm, as far as I understand, the current multi-parameter optimization algorithm like PSA takes very heavy computing and therefore a very long time to converge if you want good approximation to your predefined conditions. This gets worse when more stringent conditions and more parameters are considered. You would need a dedicated fast computer for it, which is impractical. So, 6+7 is my top concern regarding its implementation. What is your idea on this?

Thank you

Leimeng Zhuang

Reviewer #2 (Remarks to the Author):

1. The authors provide abundant and detailed information about their training algorithms and the hyperparameters, which is beneficial for readers to understand. However, the authors did not disclose the details about how to apply their training algorithms to the hardware domain. For example, the figure of the photonic circuits, the actual training parameters in experiments, the electrical circuits to control their

photonic circuits, the number of the channels, and the detailed testing procedures to implement their training algorithm are not given in their manuscript. It seems only Fig. 1d shows their control circuits. More details and figures may help the readers better understand the realization of the whole self-configuration process in experiments.

2. The authors compare non-derivative and derivative methods for the self-configuration of multipurpose waveguide mesh arrangements and programmable photonic processors. As we know, the computation complexity of evolutionary algorithms is a concern when the number of parameters we want to train is large. Could the authors evaluate the training complexity of their structure as the number of tunable components increases?

3. To derivative methods, it should be noted that we can only detect the light intensity of light beams at output ports, while the phase information can not be detected via photodetectors. Therefore, it is difficult to get the correct gradient of the loss function, which is a complex number that contains the information of both intensity and phase. According to the cost function and the equations the authors use for optimization, such as Eq. S8 and Eq. S4, S5, it seems the authors do not consider the phases of the gradient. Therefore, the derivative methods the author use may have some technical issues, which may partially explain why their optimization results based on derivative algorithms is not ideal. The authors may need to address this problem or evaluate how this problem will affect final self-configuration results.

4. In figure 2 config. 1 and config. 7, it is obvious that the FSRs are not equal. Can the authors explain the reasons? And if the target spectra/ spectral masks are ideal with equal FSP, how could you obtain the optimal results since now the cost function will be very large. In other words, if the shape of the spectra (not just loss etc) has some deviations due to some reasons such as fabrication imperfection, how could you make sure the algorithm can find the desired results?

5. A similar question. In the experimental demonstration of self-configuration based on computational optimization algorithms, "the spectral masks were obtained using previously obtained current pre-sets." The 'correct answer' already includes all the imperfection such as thermal cross talk, fabrication error etc. The optimization algorithm then only needs to regenerate the best setting, which is certainly not the real application. If we already know the configurations, we can simply store them in the processor/memory and load them when needed. The real spectral masks that should be used is the ideal one which is calculated using ideal parameters of the circuits. Then due to the imperfection of the circuits, we will obtain an optimal result which has some deviations like the shapes. In this case, probably the cost function needs to be defined in another way.

6. Programmable photonic circuits seem very attractive. However, with so many heaters to configure the circuits and the complex computing core to calculate the cost functions that requires lots of time and energy, does the FPPGA still own these advantages? What will be the limitation/ bottleneck and how can it improve to really serve in real applications?

Reviewer #3 (Remarks to the Author):

The present manuscript by Daniel Pérez López and coworkers discusses systematically the methods for self-configuration of field programmable photonic circuits, which is a very timely topic. The wide range of accessible configurations and functionality with advanced optimizations shows promise for future high-density reconfigurable photonics integration. Some theoretical portions of this manuscript have been reported in recent publications, and it is good to finally see a comprehensive report on both the theory and experiment coming to fruition – and into print soon. In general, the fit to Nature Communications seems to be appropriate. The publication is recommended, but with appropriate revisions as suggested in the following comments.

1. The total operation time for executing the computational optimization is one of the most important performance metrics. More calculation and discussions should be done to prove that in reality, the

configuration time is acceptable. It would be useful to show what the elements are that contribute to the total amount of time and how long will each of them take. According to the SI, for the current experiment settings, it will take around two hours, and it is mostly limited by the laser sweeping time. What about taking better laser sources available in the market? What are the estimated operation time for different algorithms, and for different functions (such as splitter, filter, etc.)?

2. To perfectly reach complicated target function, it can be imagined that a larger amount of devices would be needed. The authors talk about large scale integration, but they haven't done that experimentally, or computationally. Any potential problems with scaling the network would be discussed. How does the optimization time scale with the network complexity? What would be the difficulties for each optimization method to work in a large scale?

3. Cost function in equation (2). It seems to me that the equation and the explanation below are not clear enough to show the all-cross target. I assume H_{chs} should be the target channel's intensity. However, the channel that has the maximum absolute value of intensity doesn't have to be the right channel. This part needs clarification.

4. Cost function in equation (3). The authors should talk about how to determine the mask function. Specifically, is there any limitations on how arbitrary the function can be? For example, does the FSR need to be equal to certain values depending on the physical size of the network cells, as in the example in the SI? My understanding is that not all mask functions would be perfectly fitted (such as a single band-pass filter, or a resonance function with arbitrary FSR, or high-Q resonance spectra), so it could lead to the question of to what extent a function can be performed only with the mesh network, not involving HPBs.

5. Recently machine learning has been widely used for photonics design and optimization. In this case, one can imagine that if machine learning is incorporated and the system has been properly trained, the configuration of each TBU could be solved quickly. Thus it can avoid performing time-consuming computational optimizations every time for a specific function. It would be very useful if the authors can talk about this aspect and potentially compare the two methods.

REVIEWER COMMENTS

Reviewer #1 (Remarks to the Author):

Programmable photonic circuits are a highly interesting topic in the field of photonics, particular for signal processing. It promises to bring implementation of signal processors to a new generation by combining digital electronics and photonics. This practice explores the capabilities of photonics towards the complexity level of CPUs which is currently a great challenge in science and it is expected to benefit a variety of applications, e.g. AI, 5G, datacenter, etc., with significant improvement in signal processing bandwidth, latency, and power consumption which are considered the key features for next generation commercial signal processors. Therefore, the effort in this field is of high value for both academia and industry. **The work shows in this paper is an important step in the path of implementing such signal processors by addressing the challenge of automatic control for scalable circuit complexity. I agree that the results here verify authors' concept and are interesting for the general readers, particularly in terms of control and optimization algorithms, encouraging deeper study and discussions in the field. I recommend acceptance for publication,** however, I suggest the authors providing more discussion on a few points, if alright.

We appreciate the quality and constructive feedback provided by Dr. Zhuang's review. We fully agree that the outcome of this paper will encourage a deeper study of the programming strategies of programmable photonic integrated circuits. In the following points, we provide more thorough discussions related to the reviewer's suggestions and comments. Direct replies to his comments are quite extensive and unfortunately they cannot be included in full in the manuscript due to word-count restrictions. However, we have tried to include a summarized selection of the key points at certain points of the manuscript.

1. The photonic circuit in demonstration measures 300mm². When you further scale it up for higher circuit complexity. How do you deal with the uniformity and risk of defect in fabrication from a yield perspective? Would the expected yield still justify its use, compared with mature digital processors in a multi-core/parallel scheme?

As mentioned by the reviewer, the demonstration in the paper is realized for a 300mm²-size photonic integrated circuit. Part of its available area (35% approx..) is dedicated to the realization of test structures (single-cell, waveguide propagation loss, single programmable unit cells characterization, etc). Even with this moderate integration density, we have measured non-ideal effects, such as: non-uniform chip-fiber coupling loss, non-uniform Unit Cell loss, and a severe thermal crosstalk due to the lack of optimization of the heaters and the use of the same metal layer for routing and heating¹.

As mentioned by the reviewer, when one further scales it up for higher circuit complexity one would experience an increment of the impact of these non-ideal effects, in particular:

- With a direct area-cost relation, scaling up circuits translates into the need of increasing the integration density. A direct consequence of this fact is the **increment of the thermal crosstalk**, coupling together the impact and effects of every phase actuator in the circuit.
- The high refractive index contrast of a silicon waveguide that makes it possible to confine light in a small volume, makes its behavior also very sensitive to small imperfections. **Nanometer-scale geometry variations can already affect the circuit performance, limiting the scale of integration.** These are mainly arising due to

¹ D. Pérez, et al., "Multipurpose silicon photonics signal processor core," Nat. Comms. vol. 8, no. 636 (2017)

variations of the silicon thickness at the wafer-level and from deviations during the waveguide width patterning. On one hand, sidewall roughness can give rise to backscattering inside the waveguide, resulting in unwanted transmission fluctuations^{2,3}. On the other side, small deviations of a few nanometers can lead to undesired changes in the light propagation properties, originating undesired phase deviations. This impacts over the performance of components like beam splitters, which are fundamental blocks of the programmable unit cells.

From a hardware perspective, several structures and mitigation techniques have been proposed and demonstrated as standalone components to reduce the impact of uniformity and risk of defects, and tuning crosstalk. For example, it has been demonstrated that the use of adiabatic directional couplers leads to better tolerances to waveguide geometry deviations⁴ and that the use of ridge waveguide directional couplers with optimum geometry can cancel the effects due to width, gap and thickness deviations⁵. As far as thermal crosstalk is concerned, some demonstrations applied additional deep lateral air trenches⁶ and optimized the architectural PIC design to facilitate the use of thermal crosstalk cancellation techniques⁷.

However, although all of these non-ideal deviations can be mitigated, **perfect fabrication and dynamic behavior does not exist**. Even though **one or few localized defects can ruin an application specific fabrication run, waveguide mesh arrangements arises as a solution as they can be employed to optimize the circuit topology to make use of the programmable unit cells and areas of the fabricated circuit providing the best operation conditions**.

In this paper, we have demonstrated that when the waveguide mesh arrangement suffers from non-ideal effects like extra and un-evenly distributed insertion loss or severe random thermal / optical crosstalk, **the automated configuration is able to find the best circuit settings**, and selects the best programmable cells to perform a certain application. In addition, the thermal crosstalk suffered both in the simulations and the experiments here is substantially worse than the one that can be achieved with state-of-the-art.

In short, **uniformity and risk of defects in fabrication from a yield perspective is certainly an important issue** in both general-purpose and application specific large-scale circuits integrating hundreds and thousand of components. They can be mitigated employing component designs that reduce and compensate their impact. In addition, waveguide mesh arrangement can benefit from their flexibility and the proposed automated routines have demonstrated that the cooperative behavior of the system leads to self-healing functions that reduce the impact of the use of non-ideal fabrication and packaging processes automatically.

Actions:

In the discussion section we have highlighted the importance of addressing several issues limiting scalability and in particular uniformity and risks of defects in fabrication.

*The former is only one of the many challenges to address in developing large-scale programmable photonic circuits. Limitations connected to fabrication defects, design deviations, passive parasitic effects, power consumption, thermal crosstalk, and robustness of phase shifters must be considered, in conjunction with others connected to reconfiguration speed and enhanced convergence of automated functions. Due to space restrictions, we specifically cover these in **Supplementary Note 10**.*

-
- ² F. Morichetti et al., "Roughness induced backscattering in optical silicon waveguides," *Phys. Rev. Lett.*, vol. 104, no. 3, pp. 1–4, 2010.
- ³ D. Pérez, J. Capmany, "Scalable analysis for arbitrary photonic integrated waveguide meshes," *Optica*, vol 6, no. 1, pp. 19-27, 2019.
- ⁴ J.V. Capenhout, "2×2silicon electro-optic switch with 110-nm bandwidth for broadband reconfigurable optical networks," *Optics Express*, vol. 17, no. 26, (2009).
- ⁵ J. C. Mikkelsen. et. al., "Dimensional variation tolerant silicon-on-insulator directional couplers", *Optics Express* vol. 22, no. 3, pp. 3145-3150, (2014).
- ⁶ A. Masood et al., "Comparison of heater architectures for thermal control of silicon photonic circuits," 10th International Conference on Group IV Photonics, Seoul, 2013, pp. 83-84.
- ⁷ M. Milanizadeh, et al., "Canceling Thermal Cross-Talk Effects in Photonic Integrated Circuits," in *Journal of Lightwave Technology*, vol. 37, no. 4, pp. 1325-1332, (2019).

Due to space restrictions however, we instruct the reader for a more detailed discussion on this and the other issues to a new supplementary note (number 10), that has been produced for this revised version. In this note we include the following text:

Supplementary Note 10: Main challenges for large-scale programmable photonic circuits.

Physics and engineering limit the future scalability of programmable photonic circuits employing a very large number of photonic actuators (>100-1000), waveguide loss and back reflection (discussed in Supplementary Note 7), and heater performance hamper the evolution of the technology. The scalability analysis of waveguide mesh arrangements was covered in [16, 21]. Here we extend the discussion for each of the main relevant scalability limits:

Fabrication defects, design deviation and other passive parasitic effects:

The high refractive index contrast of a silicon waveguide that makes it possible to confine light in a small volume, makes its behavior also very sensitive to small imperfections. Nanometer-scale geometry variations can already affect the circuit performance, limiting the scale of integration. These are mainly arising due to variations of the silicon thickness at the wafer-level and from deviations during the waveguide width patterning. On one hand, sidewall roughness can give rise to backscattering inside the waveguide, resulting in unwanted transmission fluctuations [1, 17]. On the other hand, small deviations of a few nanometers can lead to undesired changes in the light propagation properties, originating undesired phase deviations. This impacts over the performance of components like beam splitters, which are fundamental blocks of the programmable unit cells.

From a pure hardware perspective, several structures and mitigation techniques have been proposed and demonstrated as standalone components to reduce the impact of uniformity and risk of defects, and tuning crosstalk. For example, it has been demonstrated that the use of adiabatic directional couplers leads to better tolerances to waveguide geometry deviations [18] and that the use of ridge waveguide directional couplers with optimum geometry can cancel the effects due to width, gap and thickness deviations [19]. As far as thermal crosstalk is concerned, some demonstrations applied additional deep lateral air trenches [20] and optimized the architectural PIC design to facilitate the use of thermal crosstalk cancellation techniques [6].

As discussed in the main document and in the Supplementary Note 7 (Self-healing effects), the automated configuration methods reported in this paper overcome and mitigate the aforementioned defects, by finding optimal paths and avoiding the interaction with deteriorated waveguides or unit cells. Future large-scale circuits will benefit from both fabrication-tolerant hardware and the self-healing attributes of the proposed configuration routines.

(...)

Supplementary References:

- [17] F. Morichetti, A. Canciamilla, C. Ferrari, M. Torregiani, A. Melloni, and M. Martinelli, "Roughness induced backscattering in optical silicon waveguides," *Phys. Rev. Lett.*, vol. 104, no. 3, pp. 1–4, 2010.
- [18] J.V. Capenhout, "2×2siliconelectro-optic switch with 110-nm bandwidth for broadband reconfigurable optical networks," *Optics Express*, vol. 17, no. 26, (2009).
- [19] J. C. Mikkelsen. et. al., "Dimensional variation tolerant silicon-on-insulator directional couplers", *Optics Express* vol. 22, no. 3, pp. 3145-3150, (2014).
- [20] A. Masood et al., "Comparison of heater architectures for thermal control of silicon photonic circuits," 10th International Conference on Group IV Photonics, Seoul, 2013, pp. 83-84.

2. The basic component performance metrics, like waveguide loss, back reflection, nonlinear effect, coupler function accuracy, heater maximum current, life time, and possible electromigration, of the current silicon technology fundamentally **limits the complexity and functions of such circuits**. This is a first of all physics level challenge before talking about applications. What is your view on this?

Thank you for raising this key point from the practical point of view. . Indeed, both physics and engineering limit the future scalability of programmable photonic circuits employing a

very large number of photonic actuators (>100-1000), waveguide loss and back reflection (discussed in point 1), and heater performance hamper the evolution of the technology. Our view on this issue is the following:

Novel phase tuning mechanisms and architectures need to be optimized and developed to achieve low power consumption, low tuning crosstalk and robust phase actuators. The heater evolution trend in silicon on insulator shows a mitigation of the thermal tuning crosstalk and a reduction of the overall electrical power of the tuning elements. Beyond being beneficial for the overall power consumption of the circuit, **it reduces the circuit complexity of the control electronics required for driving purposes. The current state of the technology is maturing to more robust driving and phase tuning actuators but we need further improvements to achieve the consolidation of the technology.**

A good example of the trends followed by thermo-optic actuators in silicon photonic is the following:

- The simplest and original architecture is limited to proof-of-concept devices that employ a metal layer for both routing and heating (100 mW/pi, 450 μm -long to avoid electromigration and short life-times¹). Note this is the one employed in this paper.
- Next, mature foundries have optimized the Joule-effect heater achieving better efficiencies and lengths through the improvement of their material qualities, processes, and thermo-optic waveguide geometries (30 mW/pi, 100 μm).
- In the last 5-8 years, additional techniques like the use of deep-trenches / isolation trenches, si-doped heaters are opening the path to better efficiencies (< 1mW/pi, 100 μm) and a reduction of the driving circuitry complexity. With better efficiencies, the temperature of the heater can be reduced. In particular electromigration limits are inversely proportional to the electrical current density and the temperature.

To further reduce the power consumption, alternative phase tuning mechanisms are currently being explored in many research centers and universities. **Together with thermo-optic effects, the technology needs to mature in terms of robustness.** In order to provide insights for the robustness of our demonstrator, we performed resistance variation tests driving the heaters for 1000 cycles of 0-pi and measured resistance variations lower than 1% (in this case the source of the variation is unknown but likely coming from the vibration of the electrical probes employed in the test due to PAD material expansion with temperature). However, we are aware that a slight increment in the heater current (beyond the 2pi, leads to irreversible defects in the structure. **Similar and deeper efforts need to be done by researchers and industry to provide data and tests of the technology maturity and readiness for industrialization and future commercialization.**

In short, with state-of-the-art thermo-optic waveguides, improved efficiency is leading to a reduction of the electrical current demands and an increment of robustness during their dynamic operation (lower current densities lead to the mitigation of the electro-migration effect). However, greater efforts are required to quantify and qualify the robustness of phase tuning technology in general.

Actions

In the discussion section we have highlighted the importance of addressing several issues limiting scalability and in particular physics and engineering limitations.

*The former is only one of the many challenges to address in developing large-scale programmable photonic circuits. Limitations connected to fabrication defects, design deviations, passive parasitic effects, power consumption, thermal crosstalk, **and robustness***

*of phase shifters must be considered, in conjunction with others connected to reconfiguration speed and enhanced convergence of automated functions. Due to space restrictions, we specifically cover these in **Supplementary Note 10**.*

Due to space restrictions however, we instruct the reader for a more detailed discussion on this and the other issues to a new supplementary note (number 10), that has been produced for this revised version. In this note we include the following text:

Supplementary Note 10: Main challenges for large-scale programmable photonic circuits.

Physics and engineering limit the future scalability of programmable photonic circuits employing a very large number of photonic actuators (>100-1000), waveguide loss and back reflection (discussed in Supplementary Note 7), and heater performance hamper the evolution of the technology. The scalability analysis of waveguide mesh arrangements was covered in [16, 21]. Here we extend the discussion for each of the main relevant scalability limits:

(...)

Power consumption, thermal crosstalk, and robustness of phase shifters:

Novel phase tuning mechanisms and architectures need to be optimized and developed to achieve low power consumption, low tuning crosstalk and robust phase actuators. The heater evolution trend in silicon on insulator shows a mitigation of the thermal tuning crosstalk and a reduction of the overall electrical power of the tuning elements. Beyond being beneficial for the overall power consumption of the circuit, it reduces the circuit complexity of the control electronics required for driving purposes. The current state of the technology is maturing to more robust driving and phase tuning actuators, but further improvements are required to achieve the consolidation of the technology. A good example of the trends followed by thermo-optic actuators in silicon photonic is the following:

- The simplest and original architecture is limited to proof-of-concept devices that employ a metal layer for both routing and heating (100 mW/pi, 450 μm -long to avoid electromigration and short life-times). Note this is the one employed in this paper, [3]
- Next, mature foundries have optimized the Joule-effect heater achieving better efficiencies and lengths through the improvement of their material qualities, processes, and thermo-optic waveguide geometries (30 mW/pi, 100 μm), [22].
- In the last 5-8 years, additional techniques like the use of deep-trenches / isolation trenches, si-doped heaters are opening the path to better efficiencies (< 1mW/pi, 100 μm) and a reduction of the driving circuitry complexity. With better efficiencies, the temperature of the heater can be reduced. In particular electromigration limits are inversely proportional to the electrical current density and the temperature [20].

To further reduce the power consumption, alternative phase tuning mechanisms are currently being explored in many research centers and universities. These include non-volatile tuning effects based on phase change materials, mems, and electro-optic effects. Together with thermo-optic effects, the technology needs to mature in terms of robustness. In order to provide insights for the robustness of our demonstrator, we performed resistance variation tests driving the heaters for 1000 cycles of 0-pi and measured resistance variations lower than 1% (in this case the source of the variation is unknown but likely coming from the vibration of the electrical probes employed in the test due to PAD material expansion with temperature). However, we are aware that a slight increment in the heater current (beyond the 2pi, leads to irreversible defects in the structure. Similar and deeper efforts need to be

done by researchers and industry to provide data and tests of the technology maturity and readiness for industrialization and future commercialization.

In short, with state-of-the-art thermo-optic waveguides, improved efficiency is leading to a reduction of the electrical current demands and an increment of robustness during their dynamic operation (lower current densities lead to the mitigation of the electro-migration effect). However, greater efforts are required to quantify and qualify the robustness of phase tuning technology in general.

3. Such circuits are able to perform signal processing with a fixed configuration at a time. Do you envision to use such circuits to performance computing with parameter updating speed at bit-rate level?

As mentioned by the reviewer, the presented circuits perform signal processing after a slot of time dedicated to configuration. The goal is to reduce the time consumed during the reconfiguration by increasing the speed and reducing the number of operations (processing, driving, and monitoring cycles).

Current configuration time could be divided as follows:

For each operation we compute the next configuration settings based on the current settings and the readout monitoring data. This task requires the manipulation of the said signals and the execution of the optimization algorithm. Next, we translate and transmit the next variables to the driving electronic circuitry. On a physical level, the response of the heater-based thermo-optic is limited to 2.2 μs , 5.6 and 65.5 μs for Ti-based heaters, silicon-doped heaters and Ti-based heaters with under-etched waveguides, respectively⁸. Finally, the readout operation is done followed by an analog to digital operation and a transmission of the data to the logic unit for the next operation.

While processing times of the algorithm depend on the size of the array of variables (number of driving signals) and the hardware employed, the overall time is in the μs regime for vectors between 10 to 1000 variables and current electronic processors performance. In contrast, a significant delay can be imposed by the transmitted data between the logic unit, the driving circuitry, and the monitoring circuitry. To avoid a bottleneck in the internal transmission of data required by each operation, different protocols can be employed. For example, USB 2.0 allows theoretic rates up to 480 Mbit/s and USB 3.0 allows theoretic rates up to 4.8 Gbit/s, that would enable transferring the data for 1000 channels (driving of phase actuators) in 100 μs and 10 μs , respectively. Alternative protocols like PCIeExpress can 3.x, 4.0, 5.0 or 6.0 provides better transfer rates up to few GB/s.

All in all, applying some margins, we could assume that the total delay of an “operation” (setting computation, driving and monitoring) can be potentially done in the 20-200 μs regime (5-50 kHz). In general, moving to MHz or GHz regime, would require the use of

⁸ M. Jacques, et al., “Optimization of thermo-optic phase-shifter design and mitigation of thermal crosstalk on the SOI platform” Optics Express, vol 27, no. 8, 2019.

alternative tuning mechanisms and the design and development of dedicated integrated electronics circuitry. Having said this in practice, with the exception of a handful of applications (most notably optical packet switching) there is not need for the reconfiguration speed to match the speed inherent in the dynamic properties of the optical signal.

Regarding the number of operations, further programming strategies and optimization methods can be employed. **In this paper we suggested the combination of auto-routing algorithms and advance optimization methods to reduce the number of variables (driving phases) to be optimized during the self-configuration process.** In addition, we are currently investigating alternative approaches to enhance the process convergence efficiency.

In short, to further increase the reconfiguration speed of programmable photonic circuits, faster tuning methods (electro-optic) would be required. However, in order to ensure the scalability of the circuits they must provide low loss (< 0.1 dB), low footprint and simple control electronics. In addition, the range of applications of programmable photonics spans applications which are either analog in nature or do not require real-time digital signal processing. In other words, achieving bit-rate reconfiguration times is not a strict requirement for a wide range of present and future applications.

Actions:

We have combined the proposed actions with next point (4).

4. We dont have optical memory at the moment and still need to use external digital memory for a complete signal processing system. The implementation of communication adds power consumption, uses E/O devices, and may also limit overall system processing speed. Considering this, how do you justify the advantage of using photonic processors?

The strongest capability of programmable photonics is for on the fly **analog signal processing** applications where the **cooperation and not the competition between photonics and electronics** brings additional benefits. In particular for:

- RF-photonic signal processing: Using the GHz precision signal processing of programmable photonic integrated circuits, radiofrequency (RF) signals can be manipulated with high fidelity to perform reconfigurable operations: add or drop multiple channels of radio, spread across an ultra-broadband frequency range, equalization, dispersion compensation, etc.
- Photonic neural networks: Linear-matrix multipliers and neuromorphic architectures are the base of hardware acceleration computation and novel computing schemes. These architectures and more complex combinations of programmable unit cells are being explored to produce photonic integrated circuits for optical/photonic computing. The reconfigurability of the circuit and the manipulation of analog signals are the key asset.
- Quantum signal processing: In line with the previous point, quantum signal processing can be performed employing a system's core based on a photonic integrated circuit that perform a reconfigurable optical linear signal processing (matrix multiplication).

- Added reconfigurability and flexibility for analog signal processing: channel equalization, multiple input multiple output signal processing, mode unscrambling and error correction in multiplexed links, interference filtering, optical beamformers, etc.

Of course, Programmable photonics may cooperate with digital electronics to enhance its power. In the cooperative spirit addressed above this means that all digital processing and memory operations will be implemented in electronics, while photonics will be employed when the properties of optics (broadband operation, low losses) are needed. For example, in hardware acceleration. It must be understood that in this case it is the optical light wave that carries the modulated digital signal the one that is analog processed.

Actions:

In the discussion section we have highlighted the importance of addressing several issues limiting scalability and in particular reconfiguration speed and power consumption.

*The former is only one of the many challenges to address in developing large-scale programmable photonic circuits. Limitations connected to fabrication defects, design deviations, passive parasitic effects, power consumption, thermal crosstalk, and robustness of phase shifters must be considered, in conjunction with others connected to **reconfiguration speed** and enhanced convergence of automated functions. Due to space restrictions, we specifically cover these in **Supplementary Note 10**.*

Due to space restrictions however, we instruct the reader for a more detailed discussion on this and the other issues to a new supplementary note (number 10), that has been produced for this revised version.

In this note the following text is added.

Supplementary Note 10: Main challenges for large-scale programmable photonic circuits.

(...)

Reconfiguration speed and enhanced convergence of automated functions:

Some of the automated configuration routines presented in this paper require a compilation time to configure the circuit for a certain functionality. Although some applications could be configured in runtime, the presented circuits perform signal processing after a slot of time dedicated to configuration. Most final applications would benefit from a reduction of the time consumed during the reconfiguration by increasing the speed and reducing the number of operations (processing, driving, and monitoring cycles).

In particular, one of the operations of the iterative configuration procedure could be divided as follows (See Supplementary Notes 1 and 4):

Figure 1 | Division of the different stages in one of the operations of the self-configuration methods proposed in this work.

For each operation one can compute the next configuration settings based on the current setting and the readout monitoring data. This task requires the manipulation of the said signals and the execution of the optimization algorithm as described in Supplementary Notes 1 and 4. Next, we translate and transmit the next variables to the driving electronic circuitry. On a physical level, the response of the heater-based thermo-optic is limited to 2.2 μs , 5.6 and 65.5 μs for Ti-based heaters, silicon-doped heaters and Ti-based heaters with under-etched waveguides, respectively [22]. Finally, the readout operation is done followed by an analog to digital operation and a transmission of the data to the logic unit for the next operation.

While processing times of the algorithm depend on the size of the array of variables (number of driving signals) and the hardware employed, the overall time is in the μs regime for vectors between 10 to 1000 variables and current electronic processors performance. In contrast, a significant delay can be imposed by the transmitted data between the logic unit, the driving circuitry, and the monitoring circuitry. To avoid a bottleneck in the internal transmission of data required by each operation, different protocols can be employed. For example, USB 2.0 allows theoretic rates up to 480 Mbit/s and USB 3.0 allows theoretic rates up to 4.8 Gbit/s, that would enable transferring the data for 1000 channels (driving of phase actuators) in 100 μs and 10 μs , respectively. Alternative protocols like PCIeExpress can 3.x, 4.0, 5.0 or 6.0 provides better transfer rates up to few GB/s.

All in all, applying some margins, we could assume that the total delay of an “operation” (setting computation, driving and monitoring) can be potentially done in the 20-200 μs regime (5-50 kHz). In general, moving to MHz or GHz regime, would require the use of alternative tuning mechanisms and the design and development of dedicated integrated electronics circuitry. Having said this in practice, with the exception of a handful of applications (most notably optical packet switching) there is not need for the reconfiguration speed to match the speed inherent in the dynamic properties of the optical signal.

(...)

Regarding the number of operations, further programming strategies and optimization methods can be employed. In this paper we suggested the combination of auto-routing algorithms and advance optimization methods to reduce the number of variables (driving phases) to be optimized during the self-configuration process. In addition, we are currently investigating alternative approaches to enhance the process convergence efficiency.

(...)

In short, to further increase the reconfiguration speed of programmable photonic circuits, faster tuning methods would be required. However, in order to ensure the scalability of the circuits they must provide low loss (< 0.1 dB), low footprint and simple control electronics. In addition, achieving bit-rate reconfiguration times is not a strict requirement for a wide range of present and future applications. The range of applications of programmable photonics spans applications which are either analog in nature or do not require real-time digital signal processing.

5. The operation of such photonic circuits would need particular driver circuits, cooling system, and may require light source as well as amplifiers. **When** would such photonic solutions have a definite advantage over the digital processor solutions in terms of speed and power consumption at the level of a stand-alone signal processing module? Is it possible to provide a side-by-side comparison?

As mentioned in prior points, programmable photonic approaches seek complementarity with electronics (they are conceived to complement, not to substitute), this means that there is little point in trying to make a comparison with digital signal processors. However, in our opinion, in the mid- and long term (3-5 years), the added benefits of programmable photonic processors **will allow and enhance the cooperation** with current digital processor solutions, rather than compete directly with them and enable novel applications demanding extra processing flexibility and large bandwidth.

Following the reviewer's comments, a fully functional circuit requires the integration and encapsulation of the system with the electronic circuitry, a thermal control module, and additional components like high-speed modulators, lasers and high-speed photodetectors. To achieve a mature technology and to demonstrate a competitive stand-alone signal processing module several scalability issues need to be matured:

- Photonic component improvement: Reliability of photonic integrated circuits need to improve to mitigate the defects arising from fabrication processes. As explained in point 1., waveguide meshes can employ the non-used or spare parts of the circuit to maintain its operation.

- Control electronics dedicated circuits: Current demonstrators mainly employ top-bench lab equipment to drive and monitor the circuits. Further advances are required to be on the same form factor level as with the photonic circuit. Some co-integration of electronics and photonics have been demonstrated, but a disaggregated evolution might be initially preferred to decouple the scalability problems of photonics and electronics.

- Photonic packaging of programmable large-scale circuits: Large scale packaging of photonic integrated circuits with hundreds/thousands electrical signals, high-count optical fiber array interconnection and thermo-optic modules are far from being an industry standard. Few demonstrations succeeded on the integration of more than 1000 phase shifters.

- The integration of high-speed modulators and photodetectors on silicon on insulator is granted by most foundry-enabled processes. However, recent trends suggest that future integration platforms are being equipped with the ability to integrated additional materials to a silicon substrate. Silicon nitride (for low-loss and non-linear effects), lithium Niobite (for non-linear effects, modulators, ...) and indium phosphide (optical gain, lasers, photodiodes, modulators, ...).

All in all, programmable photonics publications in the last years have focused on the proposals and proof-of-concepts. In parallel to further improvements in the number of cells, efficiencies, and integration densities, a second phase of research is expected providing a more **application-oriented view** of current technology that will allow the necessary comparison and benchmarking of programmable photonic technology with current solutions. **Again, we stress that we believe that rather to directly compete with current digital processing solutions, programmable photonic will be a key enabler of additional signal processing schemes and applications.**

6. About self-characterization, when the circuit complexity is high, how do you get access to each of the TBUs in the circuits and measure the properties? When you have a large number of unknowns and you would need as many equations to solve all of them. In practice, the number of TBUs in a 2D array would be much larger than the number of optical I/Os, if I understand it right. How would this work out? and would the solution still apply when the circuit complexity change again?

We want to thank again the reviewer for this interesting question, and we have fixed both the main and supplementary material to enhance the related content of the paper as specified in the following actions section.

As opposed to the last set of automated configuration methods, the first part of the paper deals with methods that require the launching of pre-characterization routines (described in the text and supplementary material 2). As briefly covered in this conference contribution⁹, is it possible to perform the pre-characterization of the calibration curves of waveguide mesh arrangements with a small hardware overhead enabling access to few optical source points (8% of the optical ports, 5% of the programmable unit cells), and few optical monitoring points (16% of the optical ports, 11% of the programmable unit cells). The iterative procedure, consists of:

- Setting a state close to the all-cross configuration (this is not a requirement but improves the process and reduces the number of iterations).
- Perform iterative optical power **minimization** of optical paths employing the phase shifters of the programmable unit cells **out** of the targeted path.
- Perform iterative optical power **maximization** of optical paths employing the phase shifters of the programmable unit cells **in** the targeted path.
- Approximate the calibration curve of each phase shifter through the measurement of at least three points of the electrical intensity- optical power curve.

This question is of such a relevance that we are currently working on the focused reporting and benchmarking of alternative automated characterization routines. In a nutshell, the automated characterization routines described in this paper require the use of optical monitoring points in most of the outputs. Next, we employ the autorouting methods reported in this paper to measure the optical power of port pairs. This data is shaped into a linear matrix and we perform the system inversion. As pointed by the reviewer, when scaling up the waveguide mesh arrangement, the number of TBU increases faster than the number of optical ports. **However, the number of optical paths (programmed optical interconnections) that we can measure to perform the linear matrix inversion also grows exponentially, and we can build up the matrix for the inversion procedure.** The variables included on the system inversion can be the loss of the fiber-chip couplers or on-chip monitors, the loss of the TBUs (after the calibration process). We are currently analyzing the impact of the circuit size on the pre-characterization efficiency and performance. Unfortunately, the length limitation of the paper prevents us from increasing the details, coverage and dept of our explanations.

Actions

A specific paragraph has been added this note, referenced at page 2 of the main document:

Supplementary Note 2: Pre-characterization routines

⁹D. Pérez, et al., “Field-Programmable Photonic Array for multipurpose microwave photonic applications,” IEEE Microwave Photonics Conference, 19185117, Ottawa, 2019

(...)

The scalability of the method might be compromised when scaling up the waveguide mesh arrangement. Although the number of TBU increases faster than the number of optical ports, the number of optical paths that we can measure to perform the linear matrix inversion also grows exponentially, enabling the creation of the matrix. The evolution of the method and its relation to the number of unit cells in the circuit and is currently under study. However, preliminary results show that circuits with 80- 200 TBUs achieve the self-characterization method but require additional time for the pair-ports matrix generation.

7. About optimization algorithm, as far as I understand, the current multi-parameter optimization algorithm like PSAQ takes very heavy computing and therefore a very long time to converge if you want good approximation to your predefined conditions. This gets worse when more stringent conditions and more parameters are considered. You would **need a dedicated fast computer for it, which is impractical. So, 6+7 is my top concern regarding its implementation.** What is your idea on this?

We concur with the reviewer regarding the concern about the computation times during the optimization and the number of operations to achieve convergence. However, according to our analysis and on-going work, the dedicated logic unit performing the computations does not require an impractical advanced hardware.

In particular, we expect that other limitations like packaging and electrical interfacing, and optical loss will limit the number of programmable unit cells or tunable basic unit to few hundreds or thousands phase shifters (<3000 phase shifters). The software-handling of 3000 variables is not a showstopper. **However, as mentioned by the reviewer, the convergence rates and performance are directly related to the number of variables included on the optimization process.** We are currently investigating further actions to mitigate and address this scalability issue:

- Explore the use and combinations of different optimization methods: There is no free lunch in search and optimization algorithms and our underpinning work suggests that the use of orchestrated global search and local search algorithms can improve future scalability and convergence rates by one order of magnitude.
- Combine the autorouting algorithm and the optimization methods. Once the autorouting algorithm or the loading of dynamic (relative positioned) preset configurations is selected, we can perform the optimization method only on the variables selected by the first phase. This will reduce the overall number of variables and thus the search space without compromising the circuit flexibility.
- Select a portion of the system where optimization will be performed (only including certain areas or distances to minimize the number of variables and thus search space. We can select the two targeted optical ports. An autorouting algorithm can then select the shortest path between ports. Next, we can perform the advanced optimization methods (to the targeted function) only optimizing the programmable unit cells that are at a certain interconnection (logical) distance from that short-path.

- Employing Principal Component Analysis (PCA) algorithms to eliminate variables (phase shifters) with negligible impact during the optimization process to progressively reduce the number of variables during the optimization process.

In short, the number of variables introduced during the optimization processes impacts on the performance and convergence rates of the system. As highlighted in the discussion section, smarter strategies will be required to enable the self-configuration of large scale meshes at reasonable computation times and resources. **As proposed there, the combination of both approaches (auto-routing, and optimization methods) can be employed to minimize the number of variables. Other techniques like alternative algorithms, combination with PCA, or dynamic variable selections can be employed to address future scalability limits.** However, we believe that in the nearest future, the strongest limits will be coming from the evolution of the photonic (optical loss) and electronic (interfacing) hardware.

Actions

In the discussion section we have highlighted the importance of addressing the efficient management of optimization algorithms and software programming.

*Equally important is the correct management of the complexity of optimization algorithms and circuit programming(...)These metrics can be improved enhancing the convergence and by combining both proposed configuration approaches to reduce the number of variables during the optimization process. Further details in this are in **Supplementary Note 10**.*

Due to space restrictions however, we instruct the reader for a more detailed discussion on this and the other issues to a new supplementary note (number 10), that has been produced for this revised version. In this note we include the following text:

(...)

Improved convergence:

Regarding the number of operations, further programming strategies and optimization methods can be employed. In this paper we suggested the combination of auto-routing algorithms and advance optimization methods to reduce the number of variables (driving phases) to be optimized during the self-configuration process. In addition, we are currently investigating alternative approaches to enhance the process convergence efficiency.

- Explore the use and combinations of different optimization methods: There is no free lunch in search and optimization algorithms and our underpinning work suggests that the use of orchestrated global search and local search algorithms can improve future scalability and convergence rates by one order of magnitude.
- Combine the autorouting algorithm and the optimization methods. Once the autorouting algorithm or the loading of dynamic (relative positioned) preset configurations is selected, we can perform the optimization method only on the variables selected by the first phase. This will reduce the overall number of variables and thus the search space without compromising the circuit flexibility.
- Select a portion of the system where optimization will be performed (only including certain areas or distances to minimize the number of variables and thus search space. We can select the two targeted optical ports. An autorouting algorithm can then select the shortest path between ports. Next, we can perform the advanced optimization methods (to the targeted

function) only optimizing the programmable unit cells that are at a certain interconnection (logical) distance from that short-path.

- Employing Principal Component Analysis (PCA) algorithms to eliminate variables (phase shifters) with negligible impact during the optimization process to progressively reduce the number of variables during the optimization process.

As a further motivation and demonstration of the previous points, with the following example we can demonstrate the impact of the number of tuning variables on the operations required to achieve convergence. Precisely, we will configure a simple 1x2 beamsplitter following a simple cost function that includes the 50:50 splitting and the ripple in the optimization. The output feature 1 and output feature 2 are also defined to monitor the performance, average power at the outputs and average ripple, respectively. Figure 48 (a) includes in red a path connecting the input with the two outputs. An autorouting example or a manual inspection of an advanced user would determine that these 10 TBUs (H6, H9, H16, H17, H15, H14, H13, H19, H23, H28) are enough (key) to build up the desired 1x2 splitter function. For the test, we perform the self-configuration method employing a PSO with the same hyperparameters as the one selected in the configuration of the optical filter example. First, we can include for the optimization process a single phase shifter for each of the selected TBUs, including 10 variables in the optimization process. We can define convergence as the point where the cost function becomes stable and the output features are around 4.2 dB and better than 0.5 dB, respectively. As illustrated in Fig. 48 (b), a fast configuration in less than 900 operations can be achieved. Next, we repeat the test including more tuning variables by employing the dual-drive configuration or by using additional unit cells. For the additional cells, we can employ sets of cells that are at a logical interconnection distance to the key path (in red), defining a set of unit cells at distance 1, 2 and 3. For comparison purposes, in Fig 48 (b), we can see that convergence is achieved for the single-drive case of 34 variables (distance 3) at iteration 1500. In this case, this application can be achieved without dual-drive mode as no-interferometry is fully required to achieve the targeted operation. If we include the dual-drive variables, we measured 1500 and 3000 approximated operations for the cases with 20 (key) and 68 (distance 3) variables, respectively. This proves the impact of the number of variables with the convergence rates.

Since these numbers can vary depending on the starting points and the passive phase offset of each unit cell, we ran a set of 30 independent examples with arbitrary phase offset distributions, including the different cases. The results are shown in Fig. 48 (c). Here we can observe an interesting behaviour. First, due to the cost function definition, the scenario where we optimize a reduced number of unit cells lead to sub-optimal results for the loss in the channel. Since we are considering 0.15-average loss per unit cell (See methods), we would expect an approximate optical power of 3 dB + 1.2 dB at the outputs. In a general case, the scenario with 10 unit cells (key) achieves faster convergence but suboptimal results (since it is not able to find alternative or improved paths). The more variables we include, the better performance we achieve on average and the slower convergence we achieve. We must highlight that the final performance is good for all cases. These results motivate the search of future methods of large scale circuits where the optimization is performed on a subset of the arrangement with reduced distance to the “key” cells selected by a basic auto-routing algorithm.

Figure 2 | Number of variables vs convergence test: (a) waveguide mesh arrangement with the targeted circuit and TBUs highlighted by distance to the fundamental circuit. (key), (b) convergence test results for 1 case, illustrating the cost function and output feature 1 (average optical power at outputs, and output feature 2 (mean ripple at outputs). (c) convergence test for 30 independent samples per case.

In short, to further increase the reconfiguration speed of programmable photonic circuits, faster tuning methods would be required. However, in order to ensure the scalability of the circuits they must provide low loss (< 0.1 dB), low footprint and simple control electronics. In addition, achieving bit-rate reconfiguration times is not a strict requirement for a wide range of present and future applications. The range of applications of programmable photonics spans applications which are either analog in nature or do not require real-time digital signal processing.

Regarding the convergence rates of the proposed methods, the number of variables introduced during the optimization processes impacts on the performance and convergence rates of the system. Smarter strategies will be required to enable the self-configuration of large scale meshes at reasonable computation times and resources. As proposed in this work, the combination of both approaches (auto-routing, and optimization methods) can be employed to minimize the number of variables. Other techniques like alternative algorithms, combination with PCA, or dynamic variable selections can be employed to address future scalability limits. However, we believe that in the nearest future, the strongest limits will be coming from the evolution of the photonic (optical loss) and electronic (interfacing) hardware.

Reviewer #2 (Remarks to the Author):

1. The authors provide abundant and detailed information about their training algorithms and the hyperparameters, which is **beneficial for readers to understand**. However, **the authors did not disclose the details about how to apply their training algorithms to the hardware domain**. For example, **the figure of the photonic circuits, the actual training parameters in experiments, the electrical circuits to control their photonic circuits, the number of the channels, and the detailed testing procedures to implement their training algorithm are not given in their manuscript**. It seems only Fig. 1d shows their control circuits. More details and figures may help the readers better understand the realization of the whole self-configuration process in experiments.

We appreciate the comments and overall evaluation of the reviewer. In this and the following points we illustrate how we have considered and addressed his/her comments. Regarding the application methodology of the training algorithms to the hardware domain, **an effort has been made to precise every setup proposed and employed for the project demonstrators both in the methodology section of the main manuscript and in the supplementary notes.** As mentioned by the reviewer, we employ a generic scheme depicted in Fig. 1d that includes the logic unit, the monitor array system, the driving array system, and the photonic integrated circuit. This scheme is also described and particularized for the optimizers employed in the paper in **Supplementary Note 1.**

Figure 3 | Optimization system diagram and their application to self-configuring performance of optical processors. CF: Cost function, v : vector defining the configuration variables of the system for the integrated actuators, PIC: Photonic integrated circuit. The full cycle define an operation.

In the precise example raised by the reviewer we tried to specify all those details as can be found in the subsection **Experimental demonstration of auto-routing and prior-knowledge-based algorithms:**

(..)

Following the scheme described in *¡Error! No se encuentra el origen de la referencia.(d)*, our measurement set-up consists of a Multichannel Electronic Driver Array (MEDA) subsystem based on a table-top multichannel current source along with an Optical Spectrum Analyser (OSA) as an optical monitor with custom routines in Python being run on a standard personal computer, thus completing the entire feedback loop along with a processing unit. We employed the setup in a 7-hexagon waveguide mesh with 30 TBUs with the circuit being reported in [17].

(...)The tuning steps are limited to jumps of 5 mA to prevent undesired overshoots, to protect the circuit and to better illustrate the dynamic response between configurations. Each step can be done in less than 1 second, limited by the hardware of the control system employed and its **USB connections** as well as the **Serial Peripheral Interface**.

and in the methods section:

The overall system is modified through the application of a set of electrical signals that modify the optical properties locally in the circuit. **This set of variables is defined by the vector v .** When dealing with a real system this vector can represent the electrical signal feeding each phase actuator or photonic actuator in general, whereas we employ phase shifts for the performance estimator that includes a model of the non-ideal performance of every component [29].

In this work we apply both stochastic and derivative optimization techniques. In all cases, the optical system response is given by the settings applied to each phase actuator (v),

In addition, we have a dedicated Supplementary Note 8: Description of the laboratory set-up to describe the laboratory set-up under use in the experiments.

Supplementary Note 8: Description of the laboratory set-up

In this subsection, we describe the laboratory set-up under use in the experiments and further reflect on the importance of choosing appropriately the cost function, this time in a realistic scenario.

Figure 40 sketches the 30-TBU waveguide mesh under use in our experiments. Apart from it, our measurement set-up consists of a Multichannel Electronic Driver Array (MEDA) subsystem based on a table-top multichannel current source along with an optical spectrum analyser (OSA) as an optical monitor with custom routines in Python being run on a standard personal computer thus completing the entire closed-feedback loop along with a processing unit.

Actions:

In order to increase the precision level and the reproducibility of the research, we have increased the precision and incorporated missing figures like the **number of channels or variables included on each experiment** all over the main and supplementary sections, and increased the precision on the setup description. In addition, we have included a figure of the Setup employed in the experiments in **Supplementary Note 8** (Fig. 40).

Figure 4 | (a) Sketch of our experimental set-up, consisting of a MEDA subsystem based on a table-top multichannel current source (supplying thirteen independent channels) along with an Optical Spectrum Analyser (OSA) as an optical monitor with custom routines in Python being run on a standard personal computer. (b) Labelled schematic of the waveguide mesh arrangement under test and a synthesized optical path traversing TBUs H3, H6, H11, H16, H22 and H26. (c) List of cost functions (CF) under use to generate the contour plots from $|H_{i,o}(v,\lambda)|^2$ and $|H_{i,o}(v)|^2$.

2.The authors compare non-derivative and derivative methods for the self-configuration of multipurpose waveguide mesh arrangements and programmable photonic processors. As we

know, **the computation complexity of evolutionary algorithms is a concern when the number of parameters we want to train is large**. Could the authors evaluate the training complexity of their structure as the number of tunable components increases?

We thank the reviewer for raising this point that has also been outlined by another reviewer and showing its importance. Regarding the scalability, we believe that other processes and factors like packaging and electrical interfacing, and optical loss will limit the number of programmable unit cells or tunable basic unit to few hundreds or thousands phase shifters (<3000 phase shifters) for the mid-term (5-10 years). The software-handling of 3000 variables is not a showstopper. **However, as mentioned by the reviewer, the convergence rates and performance are directly related to the number of variables included on the optimization process**. We are currently investigating further actions to mitigate and address this scalability issue:

- Explore the use and combinations of different optimization methods: There is no free lunch in search and optimization algorithms and our underpinning work suggest that the use of orchestrated global search and local search algorithms can improve future scalability and convergence rates by one order of magnitude.
- Combine the autorouting algorithm and the optimization methods. Once the autorouting algorithm or the loading of dynamic (relative positioned) preset configurations is selected, we can perform the optimization method only on the variables selected by the first phase. This reduce the overall number of variables and thus the search space while compromising the circuit flexibility.
- Select a portion of the system where optimization will be performed (only including certain areas or distances to minimize the number of variables and thus search space. We can select the two targeted optical ports. An autorouting algorithm can then select the shortest path between ports. Next, we can perform the advanced optimization methods (to the targeted function) only optimizing the programmable unit cells that are at a certain (logical) distance from that short-path.
- Employing Principal Component Analysis (PCA) algorithms to eliminate variables (phase shifters) with negligible impact during the optimization process to progressively reduce the number of variables during the optimization process.

In short, the number of variables introduced during the optimization processes impacts on the performance and convergence rates of the system. As highlighted in the discussion section, smarter strategies will be required to enable the self-configuration of large scale meshes at reasonable computation times and resources. **As proposed there, the combination of both approaches (auto-routing, and optimization methods) can be employed to minimize the number of variables. Other techniques like alternative algorithms, combination with PCA, or dynamic variable selections can be employed to address future scalability limits**. However, we believe that in the nearest future, the strongest limits will be coming from the evolution of the photonic (optical loss) and electronic (interfacing) hardware.

Actions

In the discussion section we have highlighted the importance of addressing several issues limiting scalability and in particular physics and engineering limitations.

*The former is only one of the many challenges to address in developing large-scale programmable photonic circuits. Limitations connected to fabrication defects, design deviations, passive parasitic effects, power consumption, thermal crosstalk, and robustness of phase shifters must be considered, in conjunction with others connected to reconfiguration speed and enhanced convergence of automated functions. Due to space restrictions, we specifically cover these in **Supplementary Note 10**.*

Due to space restrictions however, we instruct the reader for a more detailed discussion on this and the other issues to a new supplementary note (number 10), that has been produced for this revised version. This includes a simple example of a 1x2 beamsplitter that demonstrates the impact of the number of tuning variables on the operations required to achieve convergence. Of special interest to this question we have elaborated the example in section **Improved convergence**.

Supplementary Note 10: Main challenges for large-scale programmable photonic circuits.

(...)

Improved convergence:

Regarding the number of operations, further programming strategies and optimization methods can be employed. In this paper we suggested the combination of auto-routing algorithms and advance optimization methods to reduce the number of variables (driving phases) to be optimized during the self-configuration process. In addition, we are currently investigating alternative approaches to enhance the process convergence efficiency.

- Explore the use and combinations of different optimization methods: There is no free lunch in search and optimization algorithms and our underpinning work suggests that the use of orchestrated global search and local search algorithms can improve future scalability and convergence rates by one order of magnitude.
- Combine the autorouting algorithm and the optimization methods. Once the autorouting algorithm or the loading of dynamic (relative positioned) preset configurations is selected, we can perform the optimization method only on the variables selected by the first phase. This will reduce the overall number of variables and thus the search space without compromising the circuit flexibility.
- Select a portion of the system where optimization will be performed (only including certain areas or distances to minimize the number of variables and thus search space. We can select the two targeted optical ports. An autorouting algorithm can then select the shortest path between ports. Next, we can perform the advanced optimization methods (to the targeted function) only optimizing the programmable unit cells that are at a certain interconnection (logical) distance from that short-path.
- Employing Principal Component Analysis (PCA) algorithms to eliminate variables (phase shifters) with negligible impact during the optimization process to progressively reduce the number of variables during the optimization process.

As a further motivation and demonstration of the previous points, with the following example we can demonstrate the impact of the number of tuning variables on the operations required to achieve

convergence. Precisely, we will configure a simple 1x2 beamsplitter following a simple cost function that includes the 50:50 splitting and the ripple in the optimization. The output feature 1 and output feature 2 are also defined to monitor the performance, average power at the outputs and average ripple, respectively. Figure 48 (a) includes in red a path connecting the input with the two outputs. An autorouting example or a manual inspection of an advanced user would determine that these 10 TBUs (H6, H9, H16, H17, H15, H14, H13, H19, H23, H28) are enough (key) to build up the desired 1x2 splitter function. For the test, we perform the self-configuration method employing a PSO with the same hyperparameters as the one selected in the configuration of the optical filter example. First, we can include for the optimization process a single phase shifter for each of the selected TBUs, including 10 variables in the optimization process. We can define convergence as the point where the cost function becomes stable and the output features are around 4.2 dB and better than 0.5 dB, respectively. As illustrated in Fig. 48 (b), a fast configuration in less than 900 operations can be achieved. Next, we repeat the test including more tuning variables by employing the dual-drive configuration or by using additional unit cells. For the additional cells, we can employ sets of cells that are at a logical interconnection distance to the key path (in red), defining a set of unit cells at distance 1, 2 and 3. For comparison purposes, in Fig 48 (b), we can see that convergence is achieved for the single-drive case of 34 variables (distance 3) at iteration 1500. In this case, this application can be achieved without dual-drive mode as no-interferometry is fully required to achieve the targeted operation. If we include the dual-drive variables, we measured 1500 and 3000 approximated operations for the cases with 20 (key) and 68 (distance 3) variables, respectively. This proves the impact of the number of variables with the convergence rates.

Since these numbers can vary depending on the starting points and the passive phase offset of each unit cell, we ran a set of 30 independent examples with arbitrary phase offset distributions, including the different cases. The results are shown in Fig. 48 (c). Here we can observe an interesting behaviour. First, due to the cost function definition, the scenario where we optimize a reduced number of unit cells lead to sub-optimal results for the loss in the channel. Since we are considering 0.15-average loss per unit cell (See methods), we would expect an approximate optical power of 3 dB + 1.2 dB at the outputs. In a general case, the scenario with 10 unit cells (key) achieves faster convergence but suboptimal results (since it is not able to find alternative or improved paths). The more variables we include, the better performance we achieve on average and the slower convergence we achieve. We must highlight that the final performance is good for all cases. **These results motivate the search of future methods of large scale circuits where the optimization is performed on a subset of the arrangement with reduced distance to the “key” cells selected by a basic auto-routing algorithm and the search of alternative optimizers.**

Figure 5 | Number of variables vs convergence test: (a) waveguide mesh arrangement with the targeted circuit and TBUs highlighted by distance to the fundamental circuit. (key), (b) convergence test results for 1 case,

illustrating the cost function and output feature 1 (average optical power at outputs, and output feature 2 (mean ripple at outputs). (c) convergence test for 30 independent samples per case.

In short, to further increase the reconfiguration speed of programmable photonic circuits, faster tuning methods would be required. However, in order to ensure the scalability of the circuits they must provide low loss (< 0.1 dB), low footprint and simple control electronics. In addition, achieving bit-rate reconfiguration times is not a strict requirement for a wide range of present and future applications. The range of applications of programmable photonics spans applications which are either analog in nature or do not require real-time digital signal processing.

Regarding the convergence rates of the proposed methods, the number of variables introduced during the optimization processes impacts on the performance and convergence rates of the system. Smarter strategies will be required to enable the self-configuration of large scale meshes at reasonable computation times and resources. As proposed in this work, the combination of both approaches (auto-routing, and optimization methods) can be employed to minimize the number of variables. Other techniques like alternative algorithms, combination with PCA, or dynamic variable selections can be employed to address future scalability limits. However, we believe that in the nearest future, the strongest limits will be coming from the evolution of the photonic (optical loss) and electronic (interfacing) hardware.

3. To derivative methods, it should be noted that we can only detect the light intensity of light beams at output ports, while the phase information can not be detected via photodetectors. Therefore, it is difficult to get the correct gradient of the loss function, which is a complex number that contains the information of both intensity and phase. **According to the cost function and the equations the authors use for optimization, such as Eq. S8 and Eq. S4, S5, it seems the authors do not consider the phases of the gradient.** Therefore, the derivative methods the author use may have some technical issues, which may partially explain why their optimization results based on derivative algorithms is not ideal. **The authors may need to address this problem or evaluate how this problem will affect final self-configuration results.**

We thank the reviewer for this comment. As pointed by him/her, *although a real system contains amplitude and phase response, we employ the overall amplitude response in our current methods and demonstrations.* The reason is that from a hardware and physical perspective, retrieving the amplitude of the signal is straightforward employing optical signal monitors, either based on germanium photodetectors, non-invasive solutions or a tap joined to a grating coupler. **The lack of convergence of some of the derivative demonstrations can be explained by the well-known characteristic problematic of such-algorithms to escape from local-minima.** Derivative algorithms are better equipped to perform local search. This problematic behavior is exacerbated when the search landscape (number of variables) is greater and the cost function is more complex or includes noise.

Following the reviewer's reasoning, if an application like dispersion compensation, arbitrary phase-filter, or Talbot -effect signal processing targets the precise configuration of the phase we still envision two possible solutions. The first one deals with the definition of the cost function based on the estimation of the phase employing the Kramers–Kronig relations if the targeted function can be described as a minimum phase filter. The second approach consists in employing a cost function proportional to the application performance. For example, a dispersion compensation link can employ the aperture of the eye-diagram or the relation between pilot tones to get the cost function.

4. In figure 2 config. 1 and config. 7, **it is obvious that the FSRs are not equal**. Can the authors explain the reasons? And if the target spectra/ spectral masks are ideal with equal FSP, how could you obtain the optimal results since now the cost function will be very large. In other words, if the shape of the spectra (not just loss etc) has some deviations due to some reasons such as fabrication imperfection, how could you make sure the algorithm can find the desired results?

Thanks for your comment. As we tried to specify with the diagrams and color codes, **Configuration 1** relates to an optical ring resonator with a **cavity length of 10 TBUs** (H17, H23, H22, H21, H20, H15, H10, H11, H12, H13). Thus, the Free-spectral range is such that originates **5 notches** in the 0.3 nm-span shown in the measured response. In contrast, **Configuration 7** relates to an optical ring resonator with a cavity length **of 12 TBUs** (H17, H23, H22, **H16**, H11, H10, H15, H20, H21, **H16**, H12, H13), where H16 acts as a crossing. Thus, the free-spectral range is such that originates **6 notches** in the 0.3 nm-span shown in the measured response.

Regarding the second question, the iterative nature of the algorithms employed in the self-configuration routines will try to **minimize** a certain cost function and find the optimum value for the drivers' configuration. Sometimes, the targeted spectral mask is **too demanding** (see Fig 4 c, j) for the available hardware resources (and cannot be physically achievable- **there is not a configuration that will result in the perfect targeted function**). For example, as mentioned by the reviewer, severe fabrication errors in all the units, a too ambitious mask that requires a non achievable high-order filter or interference with the number of cells available or the propagation loss inside the reconfigurable optical circuit **can lead to an unreachable target**. In those cases, the algorithms applied in this work perform the optimization and offer a solution where the variables are tuned to get as close as possible to the targeted value. As explained before Fig. 4, *"We observed that some of the structures require a larger-scale mesh arrangement to provide the targeted higher-order filter masks"*

This can be also inferred from the Cost Function evolutions in Fig. 4, where some of the values get stacked before achieving the zero value in the cost function (see Fig 4 i and j).

One measure to enhance the convergence of all the algorithms proposed and in particular the derivative methods, consists in employing an initial configuration (set of variables) that are relatively close to the optimum value. This means that by employing the data from pre-characterization routines, auto-routing algorithms and coupling factors specifications from algorithms available in the literature for certain circuits¹⁰ (arbitrary filters, dispersion compensators, ...) , one could enhance the convergence, scarifying some freedom of the algorithm and adaptability.

Actions in the paper including measures to enhance convergence has been covered and included in the previous point 2.

5. A similar question. In the experimental demonstration of self-configuration based on computational optimization algorithms, "the spectral masks were obtained using previously

¹⁰ Optical Filter Design and Analysis: A Signal Processing Approach (Wiley Series in Microwave and Optical Engineering Book 57

obtained current pre-sets.” **The ‘correct answer’ already includes all the imperfection such as thermal cross talk, fabrication error etc. The optimization algorithm then only needs to regenerate the best setting, which is certainly not the real application.** If we already know the configurations, we can simply store them in the processor/memory and load them when needed. **The real spectral masks that should be used is the ideal one which is calculated using ideal parameters of the circuits.** Then due to the imperfection of the circuits, we will obtain an optimal result which has some deviations like the shapes. **In this case, probably the cost function needs to be defined in another way.**

The reviewer is correct on this point as we employed a spectral mask for the experimental demonstration, the response obtained in the past from a pre-set configuration. This target function includes an advantage, which is the information that we know that the optimal solution will exist and is achievable.

However, this was not the case for the examples covered in Fig. 4. There, we created arbitrary masks (black squares) and performed blind optimization.

For the creation of the masks in Fig. 4, we proceeded as follows:

-we considered the approximated average loss of each tunable basic unit, an information that can be approximated with a basic wafer level characterization of the propagation loss, and the 3-dB coupler loss.

-We applied some margins to the targeted Insertion loss and defined the targeted masks. The rationale behind this step is the following. If we know that the technology is imperfect, and we will obtain loss every time we go through a unit cell, and that the extinction ratio of the filters is also related to the achievable insertion loss, we specify a mask that provides some margin during the reconfiguration process.

However, as the reviewer will appreciate, the configuration of the arbitrary masks is quite challenging in terms of filter’s shape (giving the number of cavities) and thus, the filter order that can be obtained with the hardware. In this sense, as in the previous point, we know that most of the masks will not be purely accomplished as they are ideal. The configuration process automatically obtains the settings to get a mask as close as possible to the target. If the optimal solution does not exist, the automated process will find a close solution. **Thus, the simulation auto-configuration experiments shown in fig. 4 confirm statistically that even in the presence of un-even loss, and severe crosstalk (10% arbitrary crosstalk is far worse than the one obtained in the state of the art), the self-configuring will provide a solution.**

As the reviewer states, the capability of the waveguide mesh arrangement (not involving connected high-performance blocks) is limited by discretized structures of basic unit lengths. The unit cell length and the mesh interconnection topology set the rules of the interferometric structures that can be programmed.

In principle, as comprehensively detailed in this contribution¹¹, the hexagonal mesh is constrained to cavities of 6, 10, 12, 14, 16 TBUs and interferences of 2, 4, 6, 8, .. TBUs. **In addition, other structures like delay lines, phase shifters, Sagnac loops, multiport interferometers, can be programmed as well.**

For these examples of Fig.4, we first set the FSR of the filter from the ones available. Then we applied a sharpness factor: (10, 20, 60, 80, ...)

¹¹ Daniel Pérez, Ivana Gasulla, José Capmany, and Richard A. Soref, "Reconfigurable lattice mesh designs for programmable photonic processors," Opt. Express 24, 12093-12106 (2016)

Finally, we cut and force the limits for the stopband and passband.
The tuning of the spectra can be achieved with the value *tuning*.

```

Npoints = 21, 41, 101
FSR= 2, 4, 6, 8, 10, 12, 14, ...
Sharpnessfactor= 10, 20, 40, 80,...
IL = 3;
dB_bottom_lim = 15, 20, 30, ...
dB_bottom_lim_value = 15, 20, 30, ...
dB_top_lim = 3, 4, 5...
dB_top_lim_value = 3, 4, 5...

wn=2*pi*linspace(-0.3,0.3,Npoints)/pi;
masklin=abs(cos(wn*pi/2*FSR+tuning*pi));
mask=sharpnessfactor*log10(masklin)-IL;
mask(mask<-dB_bottom_lim)= - dB_bottom_lim_value;
ma mask(mask>-dB_top_lim)= - dB_bottom_top_value;

```

Regarding the degree of arbitrary filter' shapes that can be done, it depends on the frequency region of interest and span, the hardware employed (shape and basic unit length) and the targeted application. In principle the combination of a large scale mesh with smaller TBUs can finally become an **reconfigurable box of interferences with finer resolution** and extended possibilities. However, reducing the size/length of the unit cell is a challenge that need to be addressed and that comes with additional trade-offs.

- Accumulated loss: the creation of large circuits involving a greater number of TBUs lead to more accumulated loss. This comes with the fact that the loss of the unit cell is dominated by the loss of the 3-dB couplers. If we need to program a cavity of 1 millimeter, we would require 6 TBUs with 166 um-TBUs or 20 TBUs if they measure 50 um. The second option goes through 28 directional couplers more than the first one. Even assuming 0.1 dB/coupler, we would be adding 2.8 dB more loss to our cavity, without considering additional bend-loss, phase-shifter loss (if any).
- Reducing the size/length of the TBU increase the integration density, which might find technology limits for the tuning crosstalk effect and the electrical interfacing of such a high-number of electrical connections.

Comparative of maximum achievable Free Spectral Ranges for different waveguide mesh topologies assuming a silicon on insulator waveguide with 4.18 group index

All in all, with the current state-of-the-art, the reconfigurable optical core based on waveguide mesh arrangements limits the circuits to discretized values determined by the length of the tunable basic unit. Increasing the resolution and reducing this limit, will require the miniaturization of the unit cell. With current state of the art, it can be achieved a reduction to around 100 um, employing suspended heaters of 30 um, ultra-short 3-dB couplers of 15 um and small transitions and access waveguides. To the best of our knowledge, these had not been reported yet and represent a future technology challenge.

The second key challenge that appears due to the minimization is the tuning crosstalk. In this paper, we demonstrated that even a severe 10% **arbitrary** tuning crosstalk can be addressed with the self-configuration methods proposed.

However, we agree with the reviewer that a similar approach must be followed for the experimental example. To confirm this behaviour experimentally, we included an additional example with an arbitrary target mask (see new figure 47 in Supplementary Note 9).

Actions

We have included an additional figure (figure 47) in Supplementary Note 9 illustrating a new example using a self-defined spectral mask as input rather than an experimental one obtained from current presets such as in previous cases (figures 44-46). Also, we added the following paragraph shortly beforehand:

Finally, we synthesized the 4-TBU MZI using the set of TBUs highlighted in yellow illustrated in Figure 47. Unlike with previous experiments, this time we employed a self-made spectral mask in the same manner than with our simulations using the following expression:

$$H(\lambda) = 20 \log_{10} \left(\left| \cos \left(\pi \frac{\lambda}{FSR} - \delta \right) \right| \right) \quad (S10)$$

in which both the FSR and the wavelength λ are expressed in nm. Phase variation δ does represent any arbitrary shift of the spectral response of our filter –which as mentioned shortly backwards, can be achieved through dual drive operation–, and it is set in our case to 0 rad. In addition, we flattened both passband and eliminated band regions (-1.5 dB in the passband and -28 dB in the stopband) of our mask to provide a slightly modified spectrum from the one that can be actually achieved using the set of TBUs at our disposal. Such flat spectral response would resemble more to the one provided by a higher-order filter rather than to the one supplied by our first-order MZI. In any case, we can observe from the figure how the algorithm ‘does its best’ to match the filter response to the spectral mask provided by the end user as much as it can. In accordance to simulated results from Supplementary Note 4, the degree of similarity between both spectra is expected to increase if more electrical channels to drive a larger number of cell units are at our disposal. We performed this measurement on a different chip than the one used in figures 44-46, whose grating design was centered at 1570 nm rather than at 1585 nm. Looking at the variation of the measured MSE (Fig. 47c), it can be observed that it does not vary quite significantly compared to those from figs. 44-46. This happens due to the use of a larger number of sampling points during the experiment (501, in contrast to the 101 and 301 used in figures 44-45 and in figure 46 respectively). As a result, there is a much larger number of points close to the spectrum passband that contribute to reduce the average error, especially at early stages of the algorithm with the filter notches have not still been formed. Fig. 47(d), which illustrates the variation of the CF value with the number of operations, provides in this case a better insight of the performance of the algorithm under this scenario.

(Figures 44, 45 and 46)

Figure 6 | Experimental results of the synthesis of a 4-TBU imbalanced MZI in our waveguide mesh using PSO algorithm. (a) Schematic of the 30-TBU waveguide mesh. TBUs under use for the experiment appear highlighted in yellow and purple (dual-drive configuration), (b) Final experimental traces obtained after executing the algorithm. The spectral mask of the filter was obtained through a self-made spectral mask using the formula described in equation S10, (c) Evolution of the average MSE provided by the algorithm with the number of operations, (d) Evolution of the CF value with the number of operations.

Accordingly, the numeration of all following figures has been displaced by one.

Notice that although the mask was too exigent for the filter, the algorithm does its best to configure the available hardware and get a response as close as possible to the targeted spectrum.

6. Programmable photonic circuits seem very attractive. However, with so many heaters to configure the circuits and the complex computing core to calculate the cost functions that requires lots of time and energy, does the FPPGA still own these advantages? What will be the limitation/ bottleneck and how can it improve to really serve in real applications?

Thanks again for bringing this interesting point for further discussion. The **current state of the photonic technology is maturing to more robust driving and phase tuning actuators but we need further improvements to achieve the consolidation of the technology.** A good example of the trends followed by thermo-optic actuators in silicon photonic is the following:

- The simplest and original architecture is limited to proof-of-concept devices that employ a metal layer for both routing and heating (100 mW/pi, 450 μm -long to avoid electromigration and short life-times¹). Note this is the one employed in this paper.
- Next, mature foundries have optimized the Joule-effect heater achieving better efficiencies and lengths through the improvement of their material qualities, processes, and thermo-optic waveguide geometries (30 mW/pi, 100 μm).
- In the last 5-8 years, additional techniques like the use of deep-trenches / isolation trenches, si-doped heaters are opening the path to better efficiencies (< 1mW/pi, 100 μm) and a reduction of the driving circuitry complexity. With better efficiencies, the temperature of the heater can be reduced. In particular electromigration limits are inversely proportional to the electrical current density and the temperature.

To further reduce the power consumption, alternative phase tuning mechanisms are currently being explored in many research centers and universities. **Together with thermo-optic effects, the technology needs to mature in terms of robustness.**

Regarding the computing core to calculate the cost functions, the complexity is relative. Any electronic micro-controller available in the market for few tens of euros is suitable for the computing performance /overhead required by the examples shown before and future evolutions requiring 1000-3000 units.

Regarding the future application of the technology, we agree with the reviewer and further developments will be required to improve the system performance to serve in a wide range of applications demanding faster reconfiguration (See Reviewer 1.3 for more details). We believe that the results of this papers open the door for the automated re-configuration of waveguide mesh arrangements. This key fact will enable the use of programmable photonic circuits to a wide range of applications. Most of them would benefit from a rapid configuration (seconds) but others might tolerate longer reconfiguration times (minutes). For example the configuration of a wide variety of filters (see Fig. 4) employing the same ports, can be exploited for optical filtering applications that do not require micro-second reconfiguration times. Those reconfiguration times, and the ones discussed at Reviewer 1.3, are compatible with the strongest capability of programmable photonic processors, which is dealing with analog signal processing applications where the **cooperation of photonics and electronics** brings additional benefits:

- RF-photonic signal processing: Using the GHz precision signal processing of programmable photonic integrated circuits, radiofrequency (RF) signals can be manipulated with high fidelity to perform reconfigurable operations: add or drop multiple channels of radio, spread across an ultra-broadband frequency range, equalization, dispersion compensation, etc.
- Photonic neural networks: Linear-matrix multipliers and neuromorphic architectures are the base of hardware acceleration computation and novel computing schemes. These architectures and more complex combinations of programmable unit cells are being explored to produce photonic integrated circuits for optical/photonic computing. The reconfigurability of the circuit and the manipulation of analog signals are the key asset.
- Quantum signal processing: In line with the previous point, quantum signal processing can be performed employing a system's core based on a photonic integrated circuit that perform a reconfigurable optical linear signal processing (matrix multiplication).
- Added reconfigurability and flexibility for analog signal processing: channel equalization, multiple input multiple output signal processing, mode unscrambling and error correction in multiplexed links, interference filtering, optical beamformers, etc.

Finally, an interesting point raised by the reviewer is related to the scalability limits and the issues to be overcome. Here, we believe that the current and near-future bottleneck will be related to the interfacing and driving of phase shifters. As soon as the number of phase shifter increase, flip-chip solutions are required and need to be compatible with proper thermal management solutions, and high-count optical fiber connection. To the best of our

knowledge, interfacing a chip with more than 1500 phase shifters has not been accomplished yet, and if that is the case, the state of that technology/capability is far from being a mature technology.

A solution of that would be the use of time-multiplexing techniques to reduce the number of drivers required to drive the phase shifters.

Together with the electrical interfacing a second limitation/bottleneck will come from the design/fabrication of low-loss unit cells. The state of the art of this units is 0.4-0.8 dB. Better and fab-tolerant 3-dB couplers with lower loss and lower-propagation loss waveguides are required to achieve 0.2 dB unit cell to enable larger-scale devices. However, achieving better metrics will be a great challenge to overcome. A possible option is the co-integration of high-performance building blocks connected to the reconfigurable optical core to provide optical gain¹².

All in all, we see that the major future bottlenecks will be coming from the accumulated optical loss and the electrical interfacing and drivers.

Actions

In the discussion section we have highlighted the importance of addressing several issues limiting scalability.

*The former is only one of the many challenges to address in developing large-scale programmable photonic circuits. Limitations connected to fabrication defects, design deviations, passive parasitic effects, power consumption, thermal crosstalk, and robustness of phase shifters must be considered, in conjunction with others connected to reconfiguration speed and enhanced convergence of automated functions. Due to space restrictions, we specifically cover these in **Supplementary Note 10**.*

Due to space restrictions however, we instruct the reader for a more detailed discussion on this and the other issues to a new supplementary note (number 10), that has been produced for this revised version. In this note we elaborate on the main challenges for large-scale programmable photonic circuits, including fabrication defects, design deviation, other parasitic effects, power consumption, thermal crosstalk and robustness of phase shifters, as well as the already covered reconfiguration speed.

Supplementary Note 10: Main challenges for large-scale programmable photonic circuits.

Physics and engineering limit the future scalability of programmable photonic circuits employing a very large number of photonic actuators (>100-1000), waveguide loss and back reflection (discussed in Supplementary Note 7), and heater performance hamper the evolution of the technology. The scalability analysis of waveguide mesh arrangements was covered in [16, 21]. Here we extend the discussion for each of the main relevant scalability limits:

Fabrication defects, design deviation and other passive parasitic effects:

The high refractive index contrast of a silicon waveguide that makes it possible to confine light in a small volume, makes its behavior also very sensitive to small imperfections. Nanometer-scale geometry variations can already affect the circuit performance, limiting the scale of integration. These are mainly arising due to variations of the silicon thickness at the wafer-level and from deviations during the waveguide width patterning. On one hand, sidewall roughness can give rise to backscattering inside

¹² Camiel Op de Beeck, Bahawal Haq, Lukas Elsinger, Agnieszka Gocalinska, Emanuele Pelucchi, Brian Corbett, Günther Roelkens, and Bart Kuyken, "Heterogeneous III-V on silicon nitride amplifiers and lasers via microtransfer printing," *Optica* **7**, 386-393 (2020)

the waveguide, resulting in unwanted transmission fluctuations [1, 17]. On the other hand, small deviations of a few nanometers can lead to undesired changes in the light propagation properties, originating undesired phase deviations. This impacts over the performance of components like beam splitters, which are fundamental blocks of the programmable unit cells.

From a pure hardware perspective, several structures and mitigation techniques have been proposed and demonstrated as standalone components to reduce the impact of uniformity and risk of defects, and tuning crosstalk. For example, it has been demonstrated that the use of adiabatic directional couplers leads to better tolerances to waveguide geometry deviations [18] and that the use of ridge waveguide directional couplers with optimum geometry can cancel the effects due to width, gap and thickness deviations [19]. As far as thermal crosstalk is concerned, some demonstrations applied additional deep lateral air trenches [20] and optimized the architectural PIC design to facilitate the use of thermal crosstalk cancellation techniques [6].

As discussed in the main document and in the Supplementary Note 7 (Self-healing effects), the automated configuration methods reported in this paper overcome and mitigate the aforementioned defects, by finding optimal paths and avoiding the interaction with deteriorated waveguides or unit cells. Future large-scale circuits will benefit from both fabrication-tolerant hardware and the self-healing attributes of the proposed configuration routines.

Power consumption, thermal crosstalk, and robustness of phase shifters:

Novel phase tuning mechanisms and architectures need to be optimized and developed to achieve low power consumption, low tuning crosstalk and robust phase actuators. The heater evolution trend in silicon on insulator shows a mitigation of the thermal tuning crosstalk and a reduction of the overall electrical power of the tuning elements. Beyond being beneficial for the overall power consumption of the circuit, it reduces the circuit complexity of the control electronics required for driving purposes. The current state of the technology is maturing to more robust driving and phase tuning actuators, but further improvements are required to achieve the consolidation of the technology. A good example of the trends followed by thermo-optic actuators in silicon photonic is the following:

- The simplest and original architecture is limited to proof-of-concept devices that employ a metal layer for both routing and heating (100 mW/pi, 450 μm -long to avoid electromigration and short life-times). Note this is the one employed in this paper, [3]
- Next, mature foundries have optimized the Joule-effect heater achieving better efficiencies and lengths through the improvement of their material qualities, processes, and thermo-optic waveguide geometries (30 mW/pi, 100 μm), [22].
- In the last 5-8 years, additional techniques like the use of deep-trenches / isolation trenches, si-doped heaters are opening the path to better efficiencies (< 1mW/pi, 100 μm) and a reduction of the driving circuitry complexity. With better efficiencies, the temperature of the heater can be reduced. In particular electromigration limits are inversely proportional to the electrical current density and the temperature [20].

To further reduce the power consumption, alternative phase tuning mechanisms are currently being explored in many research centers and universities. These include non-volatile tuning effects based on phase change materials, mems, and electro-optic effects. Together with thermo-optic effects, the technology needs to mature in terms of robustness. In order to provide insights for the robustness of our demonstrator, we performed resistance variation tests driving the heaters for 1000 cycles of 0-pi and measured resistance variations lower than 1% (in this case the source of the variation is unknown but likely coming from the vibration of the electrical probes employed in the test due to PAD material expansion with temperature). However, we are aware that a slight increment in the heater current (beyond the 2pi, leads to irreversible defects in the structure. Similar and deeper efforts need to be done by researchers and industry to provide data and tests of the technology maturity and readiness for industrialization and future commercialization.

In short, with state-of-the-art thermo-optic waveguides, improved efficiency is leading to a reduction of the electrical current demands and an increment of robustness during their dynamic operation (lower current densities lead to the mitigation of the electro-migration effect). However, greater efforts are required to quantify and qualify the robustness of phase tuning technology in general.

Reconfiguration speed and enhanced convergence of automated functions:

Reconfiguration speed:

Some of the automated configuration routines presented in this paper require a compilation time to configure the circuit for a certain functionality. Although some applications could be configured in run-time, the presented circuits perform signal processing after a slot of time dedicated to configuration. Most final applications would benefit from a reduction of the time consumed during the reconfiguration by increasing the speed and reducing the number of operations (processing, driving, and monitoring cycles).

In particular, one of the operations of the iterative configuration procedure could be divided as follows (See Supplementary Notes 1 and 4):

Figure 7 | Division of the different stages in one of the operations of the self-configuration methods proposed in this work.

For each operation one can compute the next configuration settings based on the current setting and the readout monitoring data. This task requires the manipulation of the said signals and the execution of the optimization algorithm as described in Supplementary Notes 1 and 4. Next, we translate and transmit the next variables to the driving electronic circuitry. On a physical level, the response of the heater-based thermo-optic is limited to 2.2 μs , 5.6 and 65.5 μs for Ti-based heaters, silicon-doped heaters and Ti-based heaters with under-etched waveguides, respectively [22]. Finally, the readout operation is done followed by an analog to digital operation and a transmission of the data to the logic unit for the next operation.

While processing times of the algorithm depend on the size of the array of variables (number of driving signals) and the hardware employed, the overall time is in the μs regime for vectors between 10 to 1000 variables and current electronic processors performance. In contrast, a significant delay can be imposed by the transmitted data between the logic unit, the driving circuitry, and the monitoring circuitry. To avoid a bottleneck in the internal transmission of data required by each operation, different protocols can be employed. For example, USB 2.0 allows theoretic rates up to 480 Mbit/s and USB 3.0 allows theoretic rates up to 4.8 Gbit/s, that would enable transferring the data for 1000 channels (driving of phase actuators) in 100 μs and 10 μs , respectively. Alternative protocols like PCIeExpress can 3.x, 4.0, 5.0 or 6.0 provides better transfer rates up to few GB/s.

All in all, applying some margins, we could assume that the total delay of an “operation” (setting computation, driving and monitoring) can be potentially done in the 20-200 μs regime (5-50 kHz). In general, moving to MHz or GHz regime, would require the use of alternative tuning mechanisms and the design and development of dedicated integrated electronics circuitry. Having said this in practice, with the exception of a handful of applications (most notably optical packet switching) there is not need for the reconfiguration speed to match the speed inherent in the dynamic properties of the optical signal.

(...)

Reviewer #3 (Remarks to the Author):

The present manuscript by Daniel Pérez López and coworkers discusses systematically the methods for self-configuration of field programmable photonic circuits, which is a very timely topic. The wide range of accessible configurations and functionality with advanced optimizations shows promise for future high-density reconfigurable photonics integration. Some theoretical portions of this manuscript have been reported in recent publications, **and it is good to finally see a comprehensive report on both the theory and experiment coming to fruition – and into print soon. In general, the fit to Nature Communications seems to be appropriate. The publication is recommended,** but with appropriate revisions as suggested in the following comments.

We appreciate the words of the reviewer. We have tried to address and review his/her comments and suggestions to upgrade the paper.

1. The total operation time for executing the computational optimization is one of the most important performance metrics. More calculation and discussions should be done to prove that in reality, the configuration time is acceptable. It would be useful to show what the elements are that contribute to the total amount of time and how long will each of them take. According to the SI, for the current experiment settings, it will take around two hours, and it is mostly limited by the laser sweeping time. What about taking better laser sources available in the market? What are the estimated operation time for different algorithms, and for different functions (such as splitter, filter, etc.)?

Thanks for bringing out this interesting point. As mentioned by the reviewer, the reconfiguration time is of great importance by a wide range of applications. Some applications might tolerate min or hours configuration times. For example, we can envision programmable photonic systems that are configure to perform a certain photonic integrated circuit to avoid the time-consuming application specific photonic integrated circuit development cycle and the device will be maintained fixed for a considerable slot of time.

However, other applications like (photonic neural networks, reconfigurable/adaptative filters, or dispersion compensators¹³, or reconfigurable microwave photonic systems¹) benefit from faster reconfiguration times.

As mentioned to Reviewer 1, the goal is to reduce the time consumed during the reconfiguration by **increasing the speed and reducing the number of operations (processing, driving, and monitoring cycles).**

For the speed increment, current configuration time could be divided as follows:

For each operation we compute the next configuration settings based on the current settings and the readout monitoring data. This task requires the manipulation of the said signals and

¹³ Ranzini, et al., Appl. Sci. 2019, 9, 4332; doi:10.3390/app9204332

the execution of the optimization algorithm. Next, we translate and transmit the next variables to the driving electronic circuitry. On a physical level, the response of the heater-based thermo-optic is limited to 2.2 μs , 5.6 and 65.5 μs for Ti-based heaters, silicon-doped heaters and Ti-based heaters with under-etched waveguides, respectively¹⁴. Finally, the readout operation is done followed by an analog to digital operation and a transmission of the data to the logic unit for the next operation.

While processing times of the algorithm depend on the size of the array of variables (number of driving signals) and the hardware employed, the overall time is in the μs regime for vectors between 10 to 1000 variables and current electronic processors performance. In contrast, a significant delay can be imposed by the transmitted data between the logic unit, the driving circuitry, and the monitoring circuitry. To avoid a bottleneck in the internal transmission of data required by each operation, different protocols can be employed. For example, USB 2.0 allows theoretic rates up to 480 Mbit/s and USB 3.0 allows theoretic rates up to 4.8 Gbit/s, that would enable transferring the data for 1000 channels (driving of phase actuators) in 100 μs and 10 μs , respectively. Alternative protocols like PCIeExpress can 3.x, 4.0, 5.0 or 6.0 provides better transfer rates up to few GB/s.

All in all, applying some margins, we could assume that the total delay of an “operation” (setting computation, driving, and monitoring) can be potentially done in the 20-200 μs regime (5-50 kHz). In general, **moving to MHz or GHz regime**, would require the use of **alternative tuning mechanisms and the design and development of dedicated integrated electronics circuitry.**

Regarding the number of operations, further programming strategies and optimization methods can be employed to achieve better convergence rates. **In this paper we suggested the combination of auto-routing algorithms and advance optimization methods to reduce the number of variables (driving phases) to be optimized during the self-configuration process.** In addition, we are currently investigating alternative approaches to enhance the process convergence efficiency (see next question).

In short, to further increase the reconfiguration speed of programmable photonic circuits, faster tuning methods (electro-optic) would be required. However, in order to ensure the scalability of the circuits they must provide low loss (< 0.1 dB), low footprint and simple control electronics. Regarding the convergence rate and the number of iterations, the number of variables introduced during the optimization processes impacts on the performance and convergence rates of the system. As commented on the discussion section, smarter strategies will be required to enable the self-configuration of large scale meshes. However, we believe that in the nearest future, the strongest limits will be coming from the evolution of the photonic (optical loss) and electronic (interfacing) hardware.

Actions

In the discussion section we have highlighted the importance of addressing several issues limiting scalability.

*The former is only one of the many challenges to address in developing large-scale programmable photonic circuits. Limitations connected to fabrication defects, design deviations, passive parasitic effects, power consumption, thermal crosstalk, and robustness of phase shifters must be considered, in conjunction with others connected to **reconfiguration***

¹⁴ M. Jacques, et al., “Optimization of thermo-optic phase-shifter design and mitigation of thermal crosstalk on the SOI platform” Optics Express, vol 27, no. 8, 2019.

speed and enhanced convergence of automated functions. Due to space restrictions, we specifically cover these in **Supplementary Note 10**.

Due to space restrictions however, we instruct the reader for a more detailed discussion on this and the other issues to a new supplementary note (number 10), that has been produced for this revised version. In this note we discuss *reconfiguration speed*, although additional details can be found in the *Improved convergence* section.

Supplementary Note 10: Main challenges for large-scale programmable photonic circuits.

Reconfiguration speed and enhanced convergence of automated functions:

Reconfiguration speed:

Some of the automated configuration routines presented in this paper require a compilation time to configure the circuit for a certain functionality. Although some applications could be configured in run-time, the presented circuits perform signal processing after a slot of time dedicated to configuration. Most final applications would benefit from a reduction of the time consumed during the reconfiguration by increasing the speed and reducing the number of operations (processing, driving, and monitoring cycles).

In particular, one of the operations of the iterative configuration procedure could be divided as follows (See Supplementary Notes 1 and 4):

Figure 8 | Division of the different stages in one of the operations of the self-configuration methods proposed in this work.

For each operation one can compute the next configuration settings based on the current setting and the readout monitoring data. This task requires the manipulation of the said signals and the execution of the optimization algorithm as described in Supplementary Notes 1 and 4. Next, we translate and transmit the next variables to the driving electronic circuitry. On a physical level, the response of the heater-based thermo-optic is limited to 2.2 μs , 5.6 and 65.5 μs for Ti-based heaters, silicon-doped heaters and Ti-based heaters with under-etched waveguides, respectively [22]. Finally, the readout operation is done followed by an analog to digital operation and a transmission of the data to the logic unit for the next operation.

While processing times of the algorithm depend on the size of the array of variables (number of driving signals) and the hardware employed, the overall time is in the μs regime for vectors between 10 to 1000 variables and current electronic processors performance. In contrast, a significant delay can be imposed by the transmitted data between the logic unit, the driving circuitry, and the monitoring circuitry. To avoid a bottleneck in the internal transmission of data required by each operation, different protocols can be employed. For example, USB 2.0 allows theoretic rates up to 480 Mbit/s and USB 3.0 allows theoretic rates up to 4.8 Gbit/s, that would enable transferring the data for 1000 channels (driving of phase actuators) in 100 μs and 10 μs , respectively. Alternative protocols like PCIeExpress can 3.x, 4.0, 5.0 or 6.0 provides better transfer rates up to few GB/s.

Moreover, some functions require the readout from different wavelength points (see example of optical filter). In this case, the system can employ either a tunable laser, a set of fixed lasers at different

wavelengths or a broadband source plus a tunable passband filter. In most cases, the tuning speed is similar to the tuning of the phase shifter in the programmable photonic circuit.

All in all, applying some margins, we could assume that the total delay of an “operation” (setting computation, driving and monitoring) can be potentially done in the 20-200 μs regime (5-50 kHz). In general, moving to MHz or GHz regime, would require the use of alternative tuning mechanisms and the design and development of dedicated integrated electronics circuitry. Having said this in practice, with the exception of a handful of applications (most notably optical packet switching) there is not need for the reconfiguration speed to match the speed inherent in the dynamic properties of the optical signal.

(...)

2. To perfectly reach complicated target function, it can be imagined that a larger amount of devices would be needed. The authors talk about large scale integration, but they haven't done that experimentally, or computationally. Any potential problems with scaling the network would be discussed. How does the optimization time scale with the network complexity? What would be the difficulties for each optimization method to work in a large scale?

Thanks for raising this point. Indeed, the use of larger scale circuits promise the advantage of having more unit cells to play with. For example, some of the masks targeted in Fig. 4, would require a larger-scale integration to achieve some of the arbitrary shapes that calls for high-order filters or interferences. However, the optimization methods proposed here might suffer from extra delays / convergence problems. **The rationale behind is that any new additional programmable unit cell increases the number of variables to be optimized by 2** (due to the two extra phase shifters). In what is typically known as the **curse of dimensionality**, **the addition of one variable means that we need to find the optimal solution of our problem in one extra dimension. For future large-scale circuits (more than 1000 phase shifters), we would be “trying to find a needle in a higher-dimensional haystack”.** Far from being negligible, this can cause a serious decrement of the convergence rates and even affect the accomplishment of the task. **For those problems, we are currently exploring the following solutions to reduce the number of variables considered during the optimization problem.**

- Explore the use and combinations of different optimization methods: There is no free lunch in search and optimization algorithms and our underpinning work suggest that the use of orchestrated global search and local search algorithms can improve future scalability and convergence rates by one order of magnitude.
- Combine the autorouting algorithm and the optimization methods. Once the autorouting algorithm or the loading of dynamic (relative positioned) preset configurations is selected, we can perform the optimization method only on the variables selected by the first phase. This reduce the overall number of variables and thus the search space while compromising the circuit flexibility.
- Select a portion of the system where optimization will be performed (only including certain areas or distances to minimize the number of variables and thus search space. **We can select the two targeted optical ports. An autorouting algorithm can then select the shortest path between ports. Next, we can perform the advanced optimization methods (to the targeted function) only optimizing the programmable unit cells that are at a certain (logical) distance from that short-path.**

- Employing Principal Component Analysis (PCA) algorithms to eliminate variables (phase shifters) with negligible impact during the optimization process to progressively reduce the number of variables during the optimization process.

For global-search algorithms the convergence rate is severely reduced. For local search algorithms like most derivative methods or Genetic Algorithms or PSO with low inertia / momentum variables, the result might lead to the non-convergence of the optimization process due to the stacking on local minima. As mentioned, the solution here is 1) reducing the number of variables to be optimized as much as possible, 2) perform global search algorithms (till achieving a certain performance), 3) perform local search to go faster to the global-local optimum point.

Actions:

Following the reviewer advice, we incorporated a new discussion and a new Supplementary Note 10 (referenced in the discussion section) that covers the scalability problem associated to the optimizers. We reference here the convergence vs variables issue, but a deeper discussion on the hardware scalability is also included.

Supplementary Note 10: Main challenges for large-scale programmable photonic circuits.

Physics and engineering limit the future scalability of programmable photonic circuits employing a very large number of photonic actuators (>100-1000), waveguide loss and back reflection (discussed in Supplementary Note 7), and heater performance hamper the evolution of the technology. The scalability analysis of waveguide mesh arrangements was covered in [16, 21]. Here we extend the discussion for each of the main relevant scalability limits:

(...)

Improved convergence:

Regarding the number of operations, further programming strategies and optimization methods can be employed. In this paper we suggested the combination of auto-routing algorithms and advance optimization methods to reduce the number of variables (driving phases) to be optimized during the self-configuration process. In addition, we are currently investigating alternative approaches to enhance the process convergence efficiency.

- Explore the use and combinations of different optimization methods: There is no free lunch in search and optimization algorithms and our underpinning work suggests that the use of orchestrated global search and local search algorithms can improve future scalability and convergence rates by one order of magnitude.
- Combine the autorouting algorithm and the optimization methods. Once the autorouting algorithm or the loading of dynamic (relative positioned) preset configurations is selected, we can perform the optimization method only on the variables selected by the first phase. This will reduce the overall number of variables and thus the search space without compromising the circuit flexibility.
- Select a portion of the system where optimization will be performed (only including certain areas or distances to minimize the number of variables and thus search space. We can select the two targeted optical ports. An autorouting algorithm can then select the shortest path between ports. Next, we can perform the advanced optimization methods (to the targeted function) only optimizing the programmable unit cells that are at a certain interconnection (logical) distance from that short-path.

- Employing Principal Component Analysis (PCA) algorithms to eliminate variables (phase shifters) with negligible impact during the optimization process to progressively reduce the number of variables during the optimization process.

As a further motivation and demonstration of the previous points, with the following example we can demonstrate the impact of the number of tuning variables on the operations required to achieve convergence. Precisely, we will configure a simple 1x2 beamsplitter following a simple cost function that includes the 50:50 splitting and the ripple in the optimization. The output feature 1 and output feature 2 are also defined to monitor the performance, average power at the outputs and average ripple, respectively. Figure 48 (a) includes in red a path connecting the input with the two outputs. An autorouting example or a manual inspection of an advanced user would determine that these 10 TBUs (H6, H9, H16, H17, H15, H14, H13, H19, H23, H28) are enough (key) to build up the desired 1x2 splitter function. For the test, we perform the self-configuration method employing a PSO with the same hyperparameters as the one selected in the configuration of the optical filter example. First, we can include for the optimization process a single phase shifter for each of the selected TBUs, including 10 variables in the optimization process. We can define convergence as the point where the cost function becomes stable and the output features are around 4.2 dB and better than 0.5 dB, respectively. As illustrated in Fig. 48 (b), a fast configuration in less than 900 operations can be achieved. Next, we repeat the test including more tuning variables by employing the dual-drive configuration or by using additional unit cells. For the additional cells, we can employ sets of cells that are at a logical interconnection distance to the key path (in red), defining a set of unit cells at distance 1, 2 and 3. For comparison purposes, in Fig 48 (b), we can see that convergence is achieved for the single-drive case of 34 variables (distance 3) at iteration 1500. In this case, this application can be achieved without dual-drive mode as no-interferometry is fully required to achieve the targeted operation. If we include the dual-drive variables, we measured 1500 and 3000 approximated operations for the cases with 20 (key) and 68 (distance 3) variables, respectively. This proves the impact of the number of variables with the convergence rates.

Since these numbers can vary depending on the starting points and the passive phase offset of each unit cell, we ran a set of 30 independent examples with arbitrary phase offset distributions, including the different cases. The results are shown in Fig. 48 (c). Here we can observe an interesting behaviour. First, due to the cost function definition, the scenario where we optimize a reduced number of unit cells lead to sub-optimal results for the loss in the channel. Since we are considering 0.15-average loss per unit cell (See methods), we would expect an approximate optical power of 3 dB + 1.2 dB at the outputs. In a general case, the scenario with 10 unit cells (key) achieves faster convergence but suboptimal results (since it is not able to find alternative or improved paths). The more variables we include, the better performance we achieve on average and the slower convergence we achieve. We must highlight that the final performance is good for all cases. These results motivate the search of future methods of large scale circuits where the optimization is performed on a subset of the arrangement with reduced distance to the “key” cells selected by a basic auto-routing algorithm and the search of alternative optimizers.

Figure 9 | Number of variables vs convergence test: (a) waveguide mesh arrangement with the targeted circuit and TBUs highlighted by distance to the fundamental circuit. (key), (b) convergence test results for 1 case, illustrating the cost function and output feature 1 (average optical power at outputs, and output feature 2 (mean ripple at outputs). (c) convergence test for 30 independent samples per case.

In short, to further increase the reconfiguration speed of programmable photonic circuits, faster tuning methods would be required. However, in order to ensure the scalability of the circuits they must provide low loss (< 0.1 dB), low footprint and simple control electronics. In addition, achieving bit-rate reconfiguration times is not a strict requirement for a wide range of present and future applications. The range of applications of programmable photonics spans applications which are either analog in nature or do not require real-time digital signal processing.

Regarding the convergence rates of the proposed methods, the number of variables introduced during the optimization processes impacts on the performance and convergence rates of the system. Smarter strategies will be required to enable the self-configuration of large scale meshes at reasonable computation times and resources. As proposed in this work, the combination of both approaches (auto-routing, and optimization methods) can be employed to minimize the number of variables. Other techniques like alternative algorithms, combination with PCA, or dynamic variable selections can be employed to address future scalability limits. However, we believe that in the nearest future, the strongest limits will be coming from the evolution of the photonic (optical loss) and electronic (interfacing) hardware.

3. Cost function in equation (2). It seems to me that the equation and the explanation below are not clear enough to show the all-cross target. I assume Hchs should be the target channel's intensity. However, the channel that has the maximum absolute value of intensity doesn't have to be the right channel. This part needs clarification.

Thanks for raising this issue. As mentioned by the reviewer, the Hchs relates to the optical channel intensity. We have considered that the maximization of optical power at the channels defined by the port pairs in the paragraph before the equation: 12-23, 14-21, 10-1, 8-3, 13-6, 15-4, 17-2, 19-24, and 11-16, 9-18,7-20, 5-22, lead to the all-cross configuration. If we proceed with the independent configuration of the pair-port paths, the issue commented by the reviewer can arise, i.e., we could be optimizing the interconnection between channels but not necessarily the cross-state. However, the reviewer can see that if the optical channel

of **ALL** the channels is considered **simultaneously** during the **iterative** optimization process, the resulting state of every tunable unit cell will be CROSS STATE. We consider that we can improve the writing of this section to avoid misunderstanding.

We have verified the evolution of the coupling factor of each unit cell for all the examples. For reference find the case for the gradient descent (Figure 24 of **Supplementary Note 6** (Evolution of coupling factors (Ks) versus iterations. We did the same for the Genetic and Swarm algorithms for verification purposes.

(Extraction from Figure 10 | Resulting log-file of the grid-search of the optimization process of the CF (1) and the Gradient Descent algorithm with $\alpha_{mo} = 0.5$ and $\eta = 50$. Initial spectral response of the desired and undesired optical channels, final spectral response of the desired and undesired optical channels, Cost Function value per iteration, statistical results of the driving phases per iteration, coupling factor per iteration (not employed in the optimization process), common phase of each TBU per iteration (not employed in the optimization process), and initial and final histograms of the coupling factor and common phase value of each TBU and norm of the gradient.. Complementary resulting log-file of the grid-search of the optimization process of the CF (1) and the Gradient Descent algorithm with $\alpha_{mo} = 0.5$ and $\eta = 50$ including the features related to the application. Absolute value of the scattering matrix elements related to the targeted optical channels and their evolution for each iteration, related optical power per iteration.

4. Cost function in equation (3). The authors should talk about how to determine the mask function. Specifically, is there any limitations on how arbitrary the function can be? For example, does the FSR need to be equal to certain values depending on the physical size of the network cells, as in the example in the SI? My understanding is that not all mask functions would be perfectly fitted (such as a single band-pass filter, or a resonance function with arbitrary FSR, or high-Q resonance spectra), so it **could lead to the question of to what extent a function can be performed only with the mesh network**, not involving HPBs.

Thanks for raising this point. As the reviewer states, the capability of the waveguide mesh arrangement (not involving connected HPBs) is limited by discretized structures of basic unit lengths. The unit cell length and the mesh interconnection topology set the rules of the interferometric structures that can be programmed.

In principle, as comprehensively detailed in this contribution¹⁵, the hexagonal mesh is constrained to cavities of 6, 10, 12, 14, 16 TBUs and interferences of 2, 4, 6, 8, .. TBUs. **In addition, other structures like delay lines, phase shifters, Sagnac loops, multiport interferometers, can be programmed as well .As proposed by the reviewer, employing different the FSRs would require additional high performance blocks like bragg gratings, higher-q filters and larger FSR filters.**

For these examples of Fig.4, we first set the FSR of the filter from the ones available. Then we applied a sharpness factor: (10, 20, 60, 80, ...)

Finally, we cut and force the limits for the stopband and passband.

¹⁵ Daniel Pérez, Ivana Gasulla, José Capmany, and Richard A. Soref, "Reconfigurable lattice mesh designs for programmable photonic processors," Opt. Express 24, 12093-12106 (2016)

The tuning of the spectra can be achieved with the value *tuning*.

```

Npoints = 21, 41, 101
FSR= 2, 4, 6, 8, 10, 12, 14, ...
Sharpnessfactor= 10, 20, 40, 80,...
IL = 3;
dB_bottom_lim = 15, 20, 30, ...
dB_bottom_lim_value = 15, 20, 30, ...
dB_top_lim = 3, 4, 5...
dB_top_lim_value = 3, 4, 5...

wn=2*pi* linspace(-0.3,0.3,Npoints)/pi;
masklin=abs(cos(wn*pi/2*FSR+tuning*pi));
mask=sharpnessfactor*log10(masklin)-IL;
mask(mask<-dB_bottom_lim)= - dB_bottom_lim_value;
ma mask(mask>-dB_top_lim)= - dB_bottom_top_value;

```

Regarding the degree of arbitrary filter' shapes that can be done, it depends on the frequency region of interest and span, the hardware employed (shape and basic unit length) and the targeted application. In principle the combination of a large scale mesh with smaller TBUs can finally become an **reconfigurable box of interferences with finer resolution** and extended possibilities. However, reducing the size/length of the unit cell is a challenge that need to be addressed and that comes with additional trade-offs.

- Accumulated loss: the creation of large circuits involving a greater number of TBUs lead to more accumulated loss. This comes with the fact that the loss of the unit cell is dominated by the loss of the 3-dB couplers. If we need to program a cavity of 1 millimeter, we would require 6 TBUs with 166 um-TBUs or 20 TBUs if they measure 50 um. The second option goes through 28 directional couplers more than the first one. Even assuming 0.1 dB/coupler, we would be adding 2.8 dB more loss to our cavity, without considering additional bend-loss, phase-shifter loss (if any).
- Reducing the size/length of the TBU increase the integration density, which might find technology limits for the tuning crosstalk effect and the electrical interfacing of such a high-number of electrical connections.

Comparative of maximum achievable Free Spectral Ranges for different waveguide mesh topologies assuming a silicon on insulator waveguide with 4.18 group index

All in all, with the current state-of-the-art, the reconfigurable optical core based on waveguide mesh arrangements **limits the circuits to discretized values determined by the**

length of the tunable basic unit. Increasing the resolution and reducing this limit, will require the miniaturization of the unit cell. With current state of the art, it can be achieved a reduction to around 100 um, employing suspended heaters of 30 um, ultra-short 3-dB couplers of 15 um and small transitions and access waveguides. To the best of our knowledge, these had not been reported yet and represent a future technology challenge.

The second key challenge that appears due to the minimization is the tuning crosstalk. In this paper, we demonstrated that even a severe 10% **arbitrary** tuning crosstalk can be addressed with the self-configuration methods proposed.

Actions

In the discussion section we have highlighted the importance of addressing the efficient management of optimization algorithms and software programming.

*Equally important is the correct management of the complexity of optimization algorithms and circuit programming..... Further details in this are in **Supplementary Note 10**.*

Due to space restrictions however, we instruct the reader for a more detailed discussion on this and the other issues to a new supplementary note (number 10), that has been produced for this revised version. In this note we include the following text:

Resolution of the arbitrary responses of the reconfigurable optical filter

The unit cell length and the mesh interconnection topology set the rules of the interferometric structures that can be programmed.

In principle, as comprehensively detailed elsewhere [23], the hexagonal mesh is constrained to cavities of 6, 10, 12, 14, 16 TBUs and interferences of 2, 4, 6, 8, .. TBUs. In addition, other structures like delay lines, phase shifters, Sagnac loops, multiport interferometers, can be programmed as well. Employing different the FSRs would require the use of additional high-performance blocks like bragg gratings, higher-Q filters and larger FSR filters connected to the waveguide mesh arrangement.

Thus, the degree of arbitrary filter' shapes that can be done depends on the frequency region of interest and span, the hardware employed (shape and basic unit length) and the targeted application. In principle the combination of a large-scale mesh with smaller TBUs can finally become an **reconfigurable box of interferences with finer resolution** and extended possibilities. However, reducing the size/length of the unit cell is a challenge that need to be addressed and that comes with additional trade-offs.

- Accumulated loss: the creation of large circuits involving a greater number of TBUs lead to more accumulated loss. This comes with the fact that the loss of the unit cell is dominated by the loss of the 3-dB couplers. If one need to program a cavity of 1 millimeter, we would require 6 TBUs with 166 um-TBUs or 20 TBUs if they measure 50 um. The second option goes through 28 directional couplers more than the first one. Even assuming 0.1 dB/coupler, we would be adding 2.8 dB more loss to our cavity, without considering additional bend-loss, phase-shifter loss (if any).
- Reducing the size/length of the TBU increase the integration density, which might find technology limits for the tuning crosstalk effect and the electrical interfacing of such a high-number of electrical connections.

Figure 49 illustrates the relation between the Basic Unit Length and the achievable maximum Free Spectral Ranges for the different mesh topologies, assuming a silicon on insulator technology with a group index of 4.18.

Figure 11 | Comparative of maximum achievable Free Spectral Ranges for different waveguide mesh topologies assuming a silicon on insulator waveguide with 4.18 group index

All in all, with the current state-of-the-art, the reconfigurable optical core based on waveguide mesh arrangements limits the circuits to discretized values determined by the length of the tunable basic unit. Increasing the resolution and reducing this limit, will require the miniaturization of the unit cell. With current state of the art, it can be achieved a reduction to around 100 μm , employing suspended heaters of 30 μm , ultra-short 3-dB couplers of 15 μm and small transitions and access waveguides. To the best of our knowledge, these had not been reported yet and represent a future technology challenge.

The second key challenge that appears due to the minimization is the tuning crosstalk and the electrical interfacing of the phase shifters. In this paper, we demonstrated a solution for the first issue, as that even a severe 10% **arbitrary** tuning crosstalk can be addressed with the self-configuration methods proposed. For the packaging and electrical interfacing, additional technology efforts are required and expected during the next years.

5. Recently machine learning has been widely used for photonics design and optimization. In this case, one can imagine that if machine learning is incorporated and the system has been properly trained, the configuration of each TBU could be solved quickly. Thus it can avoid performing time-consuming computational optimizations every time for a specific function. It would be very useful if the authors can talk about this aspect and potentially compare the two methods.

Thanks for this contribution to the discussion. We really believe that machine learning can be employed as a competitive solution to improve the system self-configuration process. As mentioned by the reviewer, machine learning can be employed to configure photonic systems^{13,16}. In this precise case, a neural network can be trained to emulate the behavior of the system. This datasheet can be employed to get the configuration for a desired configuration or to improve the self-configuration process proposed in this paper.

This approach is in our future-works roadmap. It would require a large preliminary work to fully address this question. Recently, we applied machine-learning to learn the behavior of simple optical components¹⁷. From this experience, we believe that training a neural network to learn the behavior of the waveguide mesh arrangement is a quite hard task due to the

¹⁶ D. Zibar, H. Wymeersch, I. Lyubomirsky “Machine Learning under the spotlight,” Nature Photonics, vol. 11, n° 834, p. 749–751, 2017.

¹⁷ A. López, “”, MSc Thesis, 2019

number of variables at the inputs (phase shifters), the intercoupled behavior of the re-circular waveguide mesh arrangements and the outputs (wavelengths, amplitude, phase, optical ports, ...) , but we are definitely interested on pushing the limits of this learning process. An intermediate option is the learning of the system behavior for the specific CostFunction to be done (reducing the dimensionality of the output layer). We hope we can explore this approach in the mid term with our collaborators in Europe to enable an appropriate benchmarking and comparison between the different methods and their combinations.

Actions:

We complemented the discussion section with the mentioning of machine learning techniques as a potential solution to improve the efficiency of the self-configuration process:

Although we have provided a first demonstration of self -configuration functionality, results can be improved in terms of convergence speed. Further research is required to identify solutions incorporating more advanced optimization routines, cost function tailoring, and combination and scheduling of exploratory and local-search methods, **and machine-learning techniques [21]**.

[21] D. Zibar, H. Wymeersch, I. Lyubomirsky “Machine Learning under the spotlight,” Nature Photonics, vol. 11, n° 834, p. 749–751, 2017.

REVIEWER COMMENTS

Reviewer #1 (Remarks to the Author):

The authors have addressed my comments and provided extensive information to support their claims. I am satisfied with their answers and recommend acceptance for publication .

Reviewer #2 (Remarks to the Author):

I appreciate the authors' detailed response, which have addressed most of my concerns. However, I have a few questions that maybe misunderstood so that I would like to bring them up again .

1. In the setup figure (Figure 40) we know that an OSA is used to monitor the output. We can also know that authors use a tunable laser to sweep the wavelength. Based on my experience, there is a redundancy here. I wonder why not just use a simpler setup such as tunable laser + a power meter or broadband laser source + OSA? Either tunable laser or OSA can tell the wavelength information and a combination of them may slow down your testing since communication with OSA should be slower than that with a simple power meter. Is there anything I missed here?

2. In my previous Comment #4, when I mentioned the FSRs are not equal, I mean the FSR in one each figure. For example. In config. 1, as shown in the figure below, the red lines are all having equal length. We can see that the distance between these notches are not equal. Can you explain why? Is this predictable or prior-known? I assume in an ideal system they should be distributed equally, which also means the Mask should have ideal and equally distributed spectra, such as S10 in your revision, which is a periodic function. As a result, in such a system, the loss function of this solution with respect to the Mask may be not optimal (minimal) due to the peak/notch misalignment. In other words, it will be difficult to find a solution. In my mind, at least, the Mask model should be revised to tolerate the deviation in the FSR.

3. What is the resistance of the heater? 200mA current (Figure 41) seems a little bit large to me if 100mW/pi is used for the thermal tuning .

Some typos:

1. The captions of Figure 488, and Figure 499 are wrong.

2. 'in which their height values are expressed by means of the cost functions defined in Figure 40(b)'.
Figure 40(b) should be Figure 40(c).

Reviewer #3 (Remarks to the Author):

After reading the authors' response to all the reviewers, it seems to me that the concerns/questions of the reviewers have been addressed/answered. The publication is recommended .

Reply to reviewers

We want to thank again the deep review of the three reviewers. We are happy to see that Reviewers 1 and 3 find their questions and comments extensively addressed and recommend the paper's publication.

Regarding Reviewer 2, we have clarified the two new questions (Points 1, and 3) and worked on the analysis and explanation of Point 2.

Reviewer #2 (Remarks to the Author):

I appreciate the authors' detailed response, which have addressed most of my concerns. However, I have a few questions that maybe misunderstood so that I would like to bring them up again.

1. In the setup figure (Figure 40) we know that an OSA is used to monitor the output. We can also know that authors use a tunable laser to sweep the wavelength. Based on my experience, there is a redundancy here. **I wonder why not just use a simpler setup such as tunable laser + a power meter or broadband laser source + OSA?** Either tunable laser or OSA can tell the wavelength information and a combination of them may slow down your testing since communication with OSA should be slower than that with a simple power meter. Is there anything I missed here?

Thanks for this input. The rationale behind this decision was the following:
Some of the structures to be measured during the experiments are periodic filters. Those filters are inherently limited to a set of discretized values of FSRs that are architecture dependent. Considering the current design, those FSRs range from 297 pm (2-Basic Unit Length) to 50 pm (10-Basic Unit Length). (See Supplementary Note 10 for a deeper explanation about the discrete paths that can be synthesized with a waveguide mesh arrangement and the fixed basic unit length(delay) limits.

The use of a **Synchronized Tunable Laser Source + OSA** allowed us to achieve a 1 pm resolution, and ensure a minimum of 50 points per period, thus resolving the spectral traces with more precision. Note that the OSA and the Laser are part of a Tunable Laser Source + OSA system, and **we employed the synchronization mode already offered** by the system.

The use of a **Tunable Laser + optical power meter** (suggested by the reviewer) can be employed to speed up the data extraction process, but we do not know precisely the actual gain in terms of time (as synchronization must be warranted and programmed). At the moment of the experiment we did not have USB-GPIB output from our PDs. We are in the process of acquiring a multichannel PD array for the direct readout of multiple photodiodes, and we will employ this technique for future works.

The use of a **broadband source + OSA** is a good alternative. However, we avoided this mode as the OSA that we have (without TLS mode enabled) featured a resolution of 10 pm (for short spans), limiting the number of points per period to 29 (in structures spanning 2-Basic Unit Lengths) and 2 points (in structures spanning 10-Basic Unit Lengths). This resolution (number of points) is clearly insufficient.

In our on-going works, we are minimizing the length of the unit cell. For example, with a shorter Basic Unit Length of 300 um, the FSRs would be close to 957pm (for a 2 BUL structure) and 191 pm (for a 10-BUL structure). In that case, the broadband source + OSA would render 95 and 19 points respectively, which might be sufficient for some applications.

All in all, we employed the Tunable Laser Source + OSA system that we have in our lab, employing the TLS mode synchronization available in the system to achieve enough resolution to resolve the spectral traces optimally. In future works we will employ an array of power meters and a tunable laser source to measure in parallel multiple optical ports and to improve reconfiguration speed and parallelization during the monitoring readouts.

Actions: We incorporated in Supplementary Note 8 a precise summary of the rationale behind the system employed:

(...) Apart from it, our measurement set-up consists of **a synchronized tunable laser source**, a Multichannel Electronic Driver Array (MEDA) subsystem based on a table-top multichannel current source along with an optical spectrum analyser (OSA) as an optical monitor with custom routines in Python being run on a standard personal computer thus completing the entire closed-feedback loop along with a processing unit. **This set-up allows us to achieve a spectral resolution of 1 pm and to ensure a minimum of 50 points per period for the shortest-FSR structure, thus resolving the spectral traces - mostly periodic filters whose FSRs are inherently limited to a set of architecture-dependent discretized values (which range in our case from 297 to 50 pm) - with more precision.**

(...)

2. In my previous Comment #4, when I mentioned the FSRs are not equal, I mean the FSR in one each figure. For example. In config. 1, as shown in the figure below, the red lines are all having equal length. We can see that the distance between these notches are not equal. Can you explain why? Is this predictable or prior-known? I assume in an ideal system they should be distributed equally, which also means the Mask should have ideal and equally distributed spectra, such as S10 in your revision, which is a periodic function. As a result, in such a system, the loss function of this solution with respect to the **Mask may be not optimal (minimal)** due to the peak/notch misalignment. In other words, **it will be difficult to find a solution. In my mind, at least, the Mask model should be revised to tolerate the deviation in the FSR.**

We want to thank the reviewer for this comment, and we would like to take the opportunity to apologize about the misunderstanding with the FSR of a certain trace. With the reviewer's comment we have been able to **further study the unexpected FSR deviation** at certain wavelengths and to **propose some measures to address this issue.**

Studying the unexpected FSR deviation:

As explained by the reviewer, when we synthesize an interferometric structure, we expect a spectral response defined by a fixed (and single) Free spectral range value, i.e. all the notches being equally distributed over the measured wavelength span. After a careful analysis we believe that the rationale behind having a variable FSR can be supported by different facts:

- **Spurious paths:** During the creation of a certain structure we could be synthesizing additional undesired paths (i.e. interferometric structures) that introduce interferences with FSR deviations. For example, in the mentioned example (ORR10) we could experience a contribution from an undesired MZI4 (TCs: H12, H22, short path: H16, long path: H11, H10, H15, H20, H21). Only a small deviation from the TCs would introduce a formation of this structure mixing different free spectral ranges. Other structures are possible as well, since we used presets (from a pre-calibration stage) and we are not re-adjusting or optimizing the architecture. Remember that in

this example we are using the auto-routing self-configuration scheme and tuning crosstalk is being neglected.

- **Group index variation:** The group index deviation in wavelength is known to produce different FSRs at different wavelengths. **However, this variation is quite soft and continuous** in wavelength and is typically appreciated in large wavelength spans. Indeed, this effect is behind the technique of measuring the group index vs wavelengths employing interferometric structures, like an unbalanced MZI. In our case, the group index of the programmed waveguides employing unit cells might be subject of variations versus wavelength since a large portion of the circuit is experiencing a change in their refractive indexes. However, again, this FSR change would be appreciated for a larger span than the one shown in the examples, and would show a softer and continuous FSR variation with wavelength. This effect could be discarded in this case with a 0.3 nm span.
- **Resolution issue:** As mentioned in point 1, when measuring interferometric structures with small FSRs, a considerable resolution is required. If not enough points are employed (measurement resolution), we might not be able to resolve the spectral trace optimally. As shown in the following figure, this scenario can lead to having a trace where the points close to the notch are shifted, producing a contribution to the FSR error. Although this error can be produced, in our examples those deviations can occur up, depending on the number of points per period and the system resolution. In our Config 1 example, we have approximately 50 points per FSR and the system resolution is 1 pm, which points to a maximum resolution related deviation of up to $\pm 2\text{pm}$ range **approximately** (note that in this case both notches of a period suffer the maximum deviation of 1 pm in different directions). Again, this is a low-probability low-impact issue for this precise example.

- **Synchronization issues** (Dynamic behavior of system vs stopping issues with the OSA.): During the experiments we employed a Laser ANDO + OSA ANDO synchronized tunable laser source system to achieve high-resolution and employ the existing synchronization commercial toolbox as motivated in Point 1. However, sometimes, during few measurements **we experienced that the OSA employed interrupts the measurement for different causes (fixing synchronization issues with the laser unit, fixing synchronization issues with the PC, liberate memory, transmit and perform internal data management, etc)**. During that time, the photonic system sometimes can behave dynamically altering the system's performance. This dynamic behavior occurs due to the **non-optimal overall thermal management** implemented for the system (Peltier-cell + Thermal Control Unit + suboptimal thermal holder that allows a residual thermal propagation between the heatsink and the chip holder) and with less impact, from the actual phase shifters and tuning crosstalk. **When the system**

continued the measurements (from the previous wavelength point and a few seconds later), the overall chip(system) temperature produced a slight shift on the spectral response, modifying locally the FSR of the notch nearby.

Although all the aforementioned possibilities occur at the same time, we believe that the last one (synchronization issues and undesired dynamic behaviour) is the one having more impact. To further study it, apart from the previous arguments, we extracted the position of the notches (analytical and measured) of the different measurements included in Fig. 2 from main text for its comparison, as represented below. We observe that, indeed, there is only a significant difference (larger than 1 pm, the resolution of the OSA) in configs. 1 and 7, as pointed out by the reviewer. In the remaining scenarios, any other difference lies below or around such number.

Fig: Distance between analytic fixed notch position (considering a fix FSR) and the measured notch position. Note that the y-axis scale is different for each graph.

Consider again that in pure auto-routing based self-configuration, we did not applied a fine final optimization, and some optical and tuning crosstalk might be introduced.

Finally, for the experimental configurations employing optimization-based self-configuration (configurations in Figure 5 of the main text and figures 44, 45 and 46 from the supplementary material), the difference between final measured results and the spectral masks used to achieve them is also hardly noticeable. Precisely, as we used 101 points and a span of 0.5 nm to represent the first two ones, their spectral resolution is therefore of 5 pm between points. In those cases, only results depicted in green in figure S44 differ in one unit in x-axis (5 pm) from those from the spectral mask at 1584.95 nm, while the rest of them --and of the

remaining experiments—there is a match between the mask-notches and the measured-notches exactly. In the third circuit, using a spectral resolution of 301 points, we found only one notch mismatch of 1.67 pm (again, one unit in x-axis) at 1585.045 nm.

Addressing the FSR variation issue:

In the last section, we saw the possible causes for a FSR variation during the measurements and the magnitude of the variations experienced in our measurements. **We believe that by using the Laser + Power Meter Array option (see point 1) in future works, we will be able to mitigate most of these issues.** However, as pointed by the reviewer and as inferred from the different variation sources analyzed in the previous section, we go along with the reviewer and some solutions could be considered:

- As proposed in the discussion, auto-routing based self-configurations (Fig. 2) would benefit from a posterior fine tuning to match the targeted functionality and further correct tuning and optical crosstalks). Since after the auto-routing we achieved a solution valid or close to the optimal working configuration, the use of local-search algorithms should provide a faster final solution.
- The optimization process **could and should repeat the measurement if a sync issue is detected.** This would alleviate the synchronization issues. Although synchronization issues are given sporadically by our current synchronized TLS-system, a future tunable laser + array of power detectors could potentially experience similar synchronization errors.
- If we assume that the FSR misalignment is coming from the *spurious path, group index variation, or resolution issue* cause, additional/different features can be incorporated to the cost function to further alleviate /optimize/minimize the deviation of the FSR. This can be done following the principles of [22], where some features like the Extinction Ratio or the IL are incorporated to the cost function. In this case, at each operation, we can post process the spectral trace to extract the Extinction Ratio of the peaks separated by a specific FSR (withing a certain margin). Solutions to mitigate similar issues and/or potentially improve the convergence of different applications can range to infinite possibilities of cost-function tailoring, where additional features are considered and added. **We expect that this work will motivate application specific experts (optical filter designers, optical beamformer designers, optical switching designers, ...) to tailor and propose alternative cost functions to achieve faster reconfigurations with higher final quality.** As a final example consider the mentioned case of the optical filter. The cost function to minimize could be represented as:

$$CF = f_{mask} + f_{ER} + f_{Ripple},$$

$$f_{mask} = c_1 \cdot m.s.e.(mask, trace),$$

Where m.s.e is the mean square error between the mask and the spectral trace, f_{ER} search for the peaks separated a certain FSR (within a margin) and maximizes their ER, and f_{Ripple} search for the peaks within the passbands and minimize the ripples. Note that every feature comes with a weight (c_1, c_2, c_3, \dots) that helps the designer to equalize or priorate some features before others. In addition, a feature considering the minimization of the sum of power at a few strategic output ports can lead to the minimization of the *spurious path* problems, as they generate undesired optical power splitting that circulates all over the mesh. Consider that the features to be incorporated also depends on the type of the filter (passband, stopband, etc, ...)

In addition, a feature that can potentially improve the convergence of the self-configuration process is the minimization of the optical power at certain residual ports. This helps to focus most of the energy to the desired output ports.

- Once the structure is close to the targeted value (as in the examples in the paper) one could additionally employ fitting techniques to the targeted analytical function to alleviate any measurement resolution related issue.

All in all, we have analyzed the possible causes of a deviated targeted function, focusing on the variable FSR issue that the reviewer highlighted. After the identification of the potential causes and their description and study, we considered the inputs from the reviewer to incorporate to the description a more complete cost function to alleviate practical issues. The improvement of the self-reconfiguration process will be three-fold. First, the control system can be improved by using a tunable laser and an array of optical power meters properly synchronized. Secondly, future works should study, propose and analyze the impact of alternative optimization methods, their scheduling and their hyperparameters on the final convergence and performance. And finally, **we expect that this work will motivate application specific experts** (optical filter designers, optical beamformer designers, optical switching designers, ...) **to tailor and propose alternative cost functions** to achieve faster reconfigurations with higher final quality, as the action suggested by the reviewer.

Actions:

We incorporated the analysis in Supplementary Note 10 (final section). In addition, as the analysis is also useful for Supplementary Note 7 and Supplementary Note 8, we placed the following text:

Check the Supplementary Note 10 for the analysis of the impact on the FSR deviations on the self-configuration process and an alternative way to define more complex cost-functions.

In Supplementary Note 10:

Addressing FSR variations in interferometric structures

When synthesizing simple periodic interferometric structures, we expect a spectral response defined by a fixed (and single) free spectral range value. This does not seem to be the case of configs. 1 and 7 achieved by auto-routing technique in Fig.2 from main text, where distances between power notches are not equal. After a careful analysis, we believe that the rationale behind FSR deviations in this and other examples can be explained on account of the following facts:

- **Accidental synthesis of spurious optical paths:** during the creation of any interferometric structure, we may be synthesizing additional undesired paths (i.e., interferometric structures) that introduce interferences with alternative FSR. Taking the synthesis of the aforementioned config. 1 as an example (ORR10), we could experience a contribution from an undesired MZI4 (TCs: H12, H22; short path: H16; long path: H11, H10, H15, H20, H21). Only a small deviation

of the coupling factor from the TCs would suffice to introduce such deviation. Other structures are possible as well, since we used presets (from a pre-calibration stage) without including any re-adjustment or optimization of the resulting structure. In the case of Fig. 2 (main text), as we are using the auto-routing self-configuration scheme, tuning crosstalk is being neglected.

- **Group index variation:** this issue is known to produce different FSRs at different wavelengths. However, such variation is quite soft and continuous in wavelength, and is typically appreciated in large wavelength spans. Indeed, this effect is behind the technique of measuring the group index vs wavelengths employing different structures such as unbalanced MZIs. In our case, the group index of the programmed waveguides employing unit cells might be subject of variations versus wavelength, since a large portion of the circuit is experiencing a change in their refractive indexes. However, again, this FSR change would be appreciated for a larger span than the one shown in the examples, and would show a softer and continuous FSR variation with wavelength. This effect could be discarded in this case with a 0.3 nm span.
- **System resolution:** As mentioned in Supplementary Note 9, measuring interferometric structures with small FSRs does require a considerable measurement resolution. If not enough points are employed, we might not be able to resolve the spectral trace optimally. As shown in the following figure, this scenario can lead to having a trace where those points closer to the notch are shifted, producing their own contribution to the FSR error. In config. 1 (main text), for example, we used approximately 50 points per FSR and the system resolution is 1 pm, which points to a maximum resolution related deviation of up to +/- 2 pm range approximately -in such case, both notches of a period would suffer a maximum deviation of 1 pm in opposite directions. Again, this is a low-probability, low-impact issue for this precise example.

Figure 51 | Example of notch mismatch due to insufficient system resolution

- **Synchronization issues:** During the experiments, we employed a Laser ANDO + OSA ANDO synchronized tunable laser source system to achieve high-resolution and employ the existing synchronization commercial toolbox as motivated in Supplementary Note 9. However, sometimes, during few measurements we experienced that the OSA employed interrupts the measurement for different causes (fixing synchronization issues with the laser unit, fixing synchronization issues with the PC, liberate memory, transmit and perform internal data management, etc). During that time, the photonic system sometimes can behave dynamically altering the system's performance. This dynamic behavior occurs due to the non-optimal overall thermal management implemented for the system (Peltier-cell + Thermal Control Unit + suboptimal thermal holder that allows a residual thermal propagation between the heatsink and the chip holder) and with less impact, from the actual phase shifters and tuning crosstalk. When the system continued the measurements, from the previous wavelength point and a

few seconds later, the overall chip temperature produced a slight shift on the spectral response, modifying locally the FSR of the notch nearby.

Although all the aforementioned possibilities can occur at the same time, we believe that the last one (synchronization issues during undesired dynamic behavior) is the one having a stronger impact in this work. To further study it, apart from previous measurements, we extracted the position of the notches (analytical and measured) of the different measurements included employing the auto-routing self-configuration (Fig. 2 main text) for its comparison, as represented below. We observe that, indeed, there is only a significant difference (larger than 1 pm, the resolution of our OSA) in configs. 1 and 7. In all remaining scenarios, any other difference lies below or around such number. Note that current examples employ information from a pre-calibrated process and neglect the behaviour of the optical and tuning crosstalks.

Figure 51 | Distances (in pm) between analytic and experimental filters' notches from Fig.2 of main text. In all cases (except for configs. 1 and 7) every notch distance lies below 1 pm, the system resolution of our OSA. Note that the y-axis has a different span for each figure.

For all experimental configurations employing optimization based self-configuration (Figure 5 main text and figures 44, 45 and 46 from this supplementary material), the difference between final results employing almost any configuration and the spectral masks used to achieve them is also hardly noticeable. Precisely, as we used 101 points and a span of 0.5 nm to represent the first two ones, their spectral resolution is therefore of 5 pm. In those cases, only results depicted in green in Figure 44 differ in one unit in x-axis (5 pm) from those from the spectral mask at 1584.95 nm, while the rest of them - and of the remaining experiments- match exactly with the mask notch point. In the third circuit, whose spectral resolution was of 301 points, we found only one notch mismatch of 1.67 pm (again, one unit in x-axis) at 1585.045 nm. Note that the last example in Fig 5 Main text corresponds to the ORR10 that showed the FSR deviation in one of its notches. In this case the result is matching the expected FSR.

To mitigate all previously mentioned events, several solutions can be provided:

- Auto-routing based self-configurations would benefit from a posterior fine tuning to match the targeted functionality and further correct tuning and optical crosstalks) or by including optical crosstalk and tuning crosstalk during the pre-characterization. Since after the auto-routing we achieved a solution valid or close to the optimal working configuration, the use of local-search algorithms should also provide a faster final solution.
- The optimization process could and should repeat the measurement if a sync issue is detected. This would alleviate the synchronization issues. Although synchronization issues are given sporadically by our current synchronized TLS-system, a future tunable laser + array of power detectors could potentially experience similar synchronization errors.
- If we assume that the FSR misalignment is coming from the *spurious path, group index variation, or resolution issue* cause, additional/different features can be incorporated to the cost function to further alleviate /optimize/minimize the deviation of the FSR. This can be done following the principles of [11], where some features like the Extinction Ratio or the IL are incorporated to the cost function. In this case, at each operation, we can post process the spectral trace to extract the Extinction Ratio of the peaks separated by a specific FSR (withing a certain margin). Solutions to mitigate similar issues and/or potentially improve the convergence of different applications can range to infinite possibilities of cost-function tailoring, where additional features are considered and added. We expect that this work will motivate application specific experts (optical filter designers, optical beamformer designers, optical switching designers, ...) to tailor and propose alternative cost functions to achieve faster reconfigurations with higher final quality. As a final example consider the mentioned case of the optical filter. The cost function to minimize could be represented as:

$$CF = f_{mask} + f_{ER} + f_{Ripple},$$

$$f_{mask} = c_1 \cdot m.s.e.(mask, trace),$$

Where m.s.e is the mean square error between the mask and the spectral trace, f_{ER} search for the peaks separated a certain FSR (within a margin) and maximizes their ER, and f_{Ripple} search for the peaks within the passbands and minimize the ripples. Note that every feature comes with a weight (c_1, c_2, c_3, \dots) that helps the designer to equalize or priorate some features before others. In addition, a feature considering the minimization of the sum of power at a few strategic output ports can lead to the minimization of the *spurious path* problems, as they generate undesired optical power splitting that circulates all over the mesh. Consider that the features to be incorporated also depends on the type of the filter (passband, stopband, etc, ...). Finally, a feature that can potentially improve the convergence of the self-configuration process is the minimization of the optical power at certain residual ports. This helps to focus most of the energy to the desired output ports.

- Once the structure is close to the targeted value (as in the examples in the paper) one could additionally employ fitting techniques to the targeted analytical function to alleviate any measurement resolution related issue.

3. What is the resistance of the heater? 200mA current (Figure 41) seems a little bit large to me if 100mW/pi is used for the thermal tuning.

The resistance of the heater was analyzed in detail in [17] and in [R1]
 R1. D. Pérez PhD Thesis "Integrated microwave photonic processors using waveguide mesh cores, 2017.

In particular, it is described as the resistance in series of the external wires, the PCB metal tracks, PIC-PCB wirebonds, the PIC metal tracks, and the heater. The overall resistance ranges between 4-10 Ohms, being the one in the heaters and the PIC metal tracks the more relevant.

Note that, as mentioned in the text and in the relevant publications, the metal layer was not optimized and the same layer is used for the PIC access tracks and the heater, resulting in a heater with poor efficiency. The dissipated electrical power in form of heat is directly proportional to the metal resistance (R_{metal}). The resistance of a straight metal layer is:

$$R_{metal} = \rho \frac{L}{W \cdot t},$$

where ρ is the metal resistivity ($\Omega \text{ m}$) and, L , W and t are the length, the width and thickness in metres of the metal layer, respectively. The heaters have a length of $466 \mu\text{m}$, a width of $10 \mu\text{m}$ and a thickness of $1.8 \mu\text{m}$.

The characterized $P\pi$ was close to $100 \text{ mW}/\pi$, as mentioned by the reviewer and the relevant publications. Considering a resistance between 4 to 6 Ohms, it results in an electrical current to π -phase and 2π phase shift of (depending on the resistance):

$$I_{\pi} = \sqrt{P/R} = 158 \text{ mA to } 129 \text{ mA}$$

$$I_{2\pi} = \sqrt{P/R} = 223 \text{ mA to } 182 \text{ mA}$$

In addition, for some unit cells, the starting passive offset of the balanced MZI is not matching the theoretical cross-state due to a phase difference between the upper and lower arm of the unit cell [R1]. This phase difference arises from nano-scale fabrication deviation during the creation of the waveguide. We measured the initial state of all the MZIs and obtained an arbitrary phase offset (ϕ_p) that, in principle, does not correlate with the unit cell orientation or their position in the die. This phase offset translates the calibration curve of the MZI and can call for phase shifts larger than π to achieve a certain coupling state. As an example, see the figure below, where a shift to the left of the calibration curve of the upper phase shifter (due to the passive phase offset), demands a larger phase shift to achieve the cross state.

All in all, as covered in the main text and the supplementary material, the state of the art of thermo-optic phase shifters is optimized and P_{π} of 10-20 mW can be obtained in most of the PDKs of foundries in Europe and Asia. Since a cooperative use of both arms (drivers) can be employed to reduce the power consumption, we set a maximum current value of 200 mA for every phase actuator in the circuit.

Actions:

We placed references to related work that already includes the characterization of the chip:

Supplementary Note 8:

In this subsection, we describe the laboratory set-up under use in the experiments and further reflect on the importance of choosing appropriately the cost function, this time in a realistic scenario.

¡Error! No se encuentra el origen de la referencia. sketches the 30-TBU waveguide mesh under use in our experiments (See details [3, 24]). (..)

[24] D. Pérez PhD Thesis, "Integrated microwave photonic processors using waveguide mesh cores", 2017.

Some typos:

1. The captions of Figure 48, and Figure 49 are wrong.
2. 'in which their height values are expressed by means of the cost functions defined in Figure 40(b)'. Figure 40(b) should be Figure 40(c).

Thanks for the identification of these typos. We have addressed them accordingly.

REVIEWERS' COMMENTS

Reviewer #2 (Remarks to the Author):

The authors have addressed my concerns. I have no more questions.